# Fine-tuning of Notch signaling sets the boundary of the organ of Corti and establishes sensory cell fates

Martin L Basch[1†], Rogers M Brown II[2], Hsin-I Jen[2], Fatih Semerci[2], Frederic Depreux[3], Renée K Edlund[2], Hongyuan Zhang[1], Christine R Norton[4], Thomas Gridley[4], Susan E Cole[5], Angelika Doetzlhofer[6], Mirjana Maletic-Savatic[1,2,7,8], Neil Segil[9], Andrew K Groves[1,2,10]*

[1]Department of Neuroscience, Baylor College of Medicine, Houston, United States; [2]Program in Developmental Biology, Baylor College of Medicine, Houston, United States; [3]Department of Cell Biology and Anatomy, Rosalind Franklin University of Medicine and Science, Chicago, United States; [4]Maine Medical Center Research Institute, Scarborough, United States; [5]Department of Molecular Genetics, The Ohio State University, Columbus, United States; [6]Solomon H. Snyder Department of Neuroscience, Johns Hopkins University, School of Medicine, Baltimore, United States; [7]Department of Pediatrics, Baylor College of Medicine, Houston, United States; [8]Jan and Dan Duncan Neurological Research Institute at Texas Children's Hospital, Houston, United States; [9]Department of Stem Cell Biology and Regenerative Medicine, Keck School of Medicine, University of Southern California, Los Angeles, United States; [10]Department of Molecular and Human Genetics, Baylor College of Medicine, Houston, United States

*For correspondence: akgroves@bcm.edu

Present address: [†]Department of Otolaryngology Head and Neck Surgery, University Hospitals, Case Medical Center, Case Western Reserve University, Cleveland, United States

Competing interests: The authors declare that no competing interests exist.

**Abstract** The signals that induce the organ of Corti and define its boundaries in the cochlea are poorly understood. We show that two Notch modifiers, *Lfng* and *Mfng,* are transiently expressed precisely at the neural boundary of the organ of Corti. Cre-Lox fate mapping shows this region gives rise to inner hair cells and their associated inner phalangeal cells. Mutation of *Lfng* and *Mfng* disrupts this boundary, producing unexpected duplications of inner hair cells and inner phalangeal cells. This phenotype is mimicked by other mouse mutants or pharmacological treatments that lower but not abolish Notch signaling. However, strong disruption of Notch signaling causes a very different result, generating many ectopic hair cells at the expense of inner phalangeal cells. Our results show that Notch signaling is finely calibrated in the cochlea to produce precisely tuned levels of signaling that first set the boundary of the organ of Corti and later regulate hair cell development.

## Introduction

The mammalian cochlear duct is a sound transducer of exquisite mechanical sensitivity. Its sensory organ, the organ of Corti, contains mechanosensory inner and outer hair cells, surrounded by different types of supporting cells (*Kelley et al., 2009*; *Basch et al., 2016*). The organ of Corti is bounded by two populations of non-sensory epithelium, the inner and outer sulci. These three domains of the cochlear duct are induced in response to a gradient of canonical BMP signaling established by the expression of BMP4 in the future outer sulcus (*Ohyama et al., 2010*). High levels of BMP signaling promote outer sulcus differentiation, low or absent BMP signaling leads to formation of the inner

sulcus, while intermediate levels of BMP signaling position the cochlear prosensory domain which will differentiate into the organ of Corti (*Ohyama et al., 2010*). Inner hair cells, which are the first cells to differentiate in the organ of Corti, arise at the boundary of the prosensory domain and the future inner sulcus, also known as Kölliker's organ, in response to inducing signals (*Groves and Fekete, 2012*). Recent work has identified several candidate signals that initiate hair cell development, including the canonical Wnt signaling pathway, that promote hair cell differentiation (*Shi et al., 2010*; *Jacques et al., 2012*; *Shi et al., 2014*; *Jansson et al., 2015*) and other signals, such as Shh, that hold hair cell differentiation in abeyance until they are down-regulated (*Bok et al., 2013*; *Tateya et al., 2013*).

A central question in understanding the mechanism of signaling gradients is how a continuously changing, 'analogue' gradient of signaling is translated into a small number of distinct, 'digital' domains, and how the boundaries between these domains are established. The Notch signaling pathway has been implicated in the establishment of boundaries between developmental territories (*Bray, 2006*; *Artavanis-Tsakonas and Muskavitch, 2010*). For example, a zone of Notch signaling is established between the dorsal and ventral halves of the *Drosophila* wing imaginal disc (*Baker, 2007*). Here, the dorsal half of the imaginal disc expresses Notch, the glycosyltransferase enzyme Fringe and the Notch ligands Delta and Serrate, while the ventral territory expresses only Delta and Notch. Fringe proteins modify Notch receptors and ligands to increase the level of Notch signaling by Delta ligands and to attenuate Notch signaling by Serrate ligands (*Rana and Haltiwanger, 2011*; *LeBon et al., 2014*). Accordingly, the action of Fringe in the wing imaginal disc serves to attenuate Serrate-Notch signaling in the dorsal region of the disc (*Rana and Haltiwanger, 2011*), but permits a sharp boundary of Notch signaling at the boundary between dorsal and ventral halves in response to Serrate and Delta signals (*Fortini, 2000*). The situation in vertebrates is complicated by the presence of multiple Delta homologues (Dll1, 3 and 4) and two Serrate homologues, Jag1 and Jag2. Current evidence suggests that Fringe modification of Notch receptors tends to *enhance* signaling by Dll1 and Dll4 ligands and *attenuate* signaling by Jag1 and Jag2 (*Hicks et al., 2000*; *LeBon et al., 2014*).

We found that two Fringe genes, *Lfng* and *Mfng*, have an extremely dynamic expression pattern in the cochlea as the progenitor cells of the organ of Corti - the so-called prosensory domain – begin to differentiate into hair cells and supporting cells. Strikingly, we find these two genes are expressed transiently precisely where inner hair cells will differentiate, at the boundary between the prosensory domain and the non-sensory region of the cochlear duct known as Kölliker's organ. *Lfng* and *Mfng* expression subsequently diverge as hair cells and their surrounding supporting cells differentiate. Our observations suggest that Notch signaling may act to first position the boundary between the future organ of Corti and Kölliker's organ, and then subsequently regulate the correct formation of inner hair cells and their neighboring supporting cells. To test this, we systematically inactivated *Lfng* and *Mfng*, Notch receptors, Notch ligands, and other regulators of the Notch pathway in the developing cochlea. We find Notch signaling controls two sets of decisions at the edge of the organ of Corti. The first decision restricts the first differentiating inner hair cells and their associated supporting cells, the inner phalangeal cells, to the boundary with Kölliker's organ. We find this fate decision is regulated by Fringe activity, requires low levels of Notch signaling and is extremely sensitive to changes in signaling strength. The second decision regulates the proportion of hair cells and supporting cells through previously characterized forms of lateral inhibition (*Lewis, 1991*, *1998*; *Kiernan, 2013*). This fate decision does not require Fringe activity, requires higher levels of Notch signaling, and is much less sensitive to small changes in signaling strength. Our results suggest that qualitatively different forms of Notch signaling regulate different fate decisions during organ of Corti development.

## Results

### Lunatic Fringe and Manic Fringe converge at the future inner hair cell region and are required to regulate inner hair cell and inner phalangeal cell differentiation

Previous studies reported that *Lunatic Fringe* (*Lfng*) is expressed in the cochlea in Kölliker's organ with the Notch ligand *Jag1* before the formation of the first inner hair cells (*Morsli et al., 1998*;

*Murata et al., 2006*; *Ohyama et al., 2010*; *Basch et al., 2011*). As the first hair cell progenitors differentiate near the base of the cochlea, they express *Atoh1* and *Manic Fringe* (*Mfng*; *Cai et al., 2013*, *2015*). To examine the changes in expression of these genes during cochlear development and hair cell differentiation, we examined the developing mouse cochlea between E13.5 and E15.5. We performed in situ hybridization for *Lfng*, *Mfng* and *Atoh1* in adjacent serial sections (*Figure 1A*) and examined Jag1 expression in *Lfng-GFP* transgenic reporter mice from the GENSAT project (*Gong et al., 2003*; *Geschwind, 2004*; *Heintz, 2004*; *Schmidt et al., 2013*) in which GFP is expressed under control of a bacterial artificial chromosome containing the *Lfng* locus. We also examined Jag1 expression in *Atoh1-GFP* knock-in mice in which GFP is fused to the coding region of *Atoh1* (*Shroyer et al., 2007*). The Atoh1-GFP fusion protein is expressed a little later than *Atoh1* mRNA (*Cai et al., 2013*), but also provides a reliable indicator of differentiating hair cells.

We observed striking dynamic changes in the expression of *Lfng*, *Mfng*, Jag1 and *Atoh1* in the cochlea between E13.5 and E15.5 as differentiation proceeds in a basal-apical direction (*Figure 1A–C*; summarized in *Figure 1D*). Jag1 is initially expressed in Kölliker's organ with Lfng at E13.5 (*Ohyama et al., 2010*), and the expression of both genes then shifts into the differentiating prosensory domain and ultimately to supporting cells over the next 48 hr (*Morsli et al., 1998*; *Murata et al., 2006*). At E14.5, *Lfng* is down-regulated in much of Kölliker's organ in the basal turn of the cochlea and is restricted to a column of differentiating Atoh1-expressing cells at the boundary of Kölliker's organ and the prosensory domain (*Figure 1A–C*). Differentiating hair cell progenitors are observed in the mid-basal prosensory domain at the boundary of Kölliker's organ starting at E13.5, where they express *Atoh1* (*Cai et al., 2013*, *2015*). Prior to E14.5, *Manic Fringe* (*Mfng*) was not detected in the cochlear duct, but at E14.5 and E15.5, it is co-expressed with Atoh1 in differentiating hair cells (*Figure 1A*; *Cai et al., 2015*).

Due to the basal-apical gradient of differentiation in the cochlea, we found that the most accurate picture of these dynamic expression changes could be obtained by examining serial sections along the length of the E15.5 cochlear duct. At this age, the apex of the cochlea contains the most immature cells, while the base of the cochlea represents the most mature differentiating hair cells and supporting cells (*Figure 1A*). We observed that *Lfng* and *Mfng* transiently co-localize at the site of *Atoh1* expression at the boundary of Kölliker's organ and the prosensory domain. We confirmed this co-localization of *Lfng* and *Mfng* by performing fluorescent in situ hybridization for *Mfng* in *Lfng-GFP* mice (*Figure 1B*). As hair cell and supporting cell differentiation continues, Fringe gene expression diverges once more, with *Lfng* and Jag1 becoming restricted to supporting cells (*Morsli et al., 1998*; *Murata et al., 2006*), and *Mfng* being restricted to hair cells. A more complete apical-basal series of serial in situ hybridization images are shown in *Figure 1—figure supplement 1*. The dynamic gene expression changes occurring along the apical-basal axis of the cochlea at E15.5 are summarized in *Figure 1D*.

The position of the transient stripe of Lfng- and Mfng-expressing cells at the border of Kölliker's organ and the prosensory domain suggested these cells give rise to inner hair cells and/or inner phalangeal cells. To test this, we performed a fate mapping experiment with Lfng-CreER transgenic mice generated with the same bacterial artificial chromosome used to produce the Lfng-GFP transgenic line . We mated Lfng-CreER transgenic mice with Ai3 ROSA Cre reporter mice (*Madisen et al., 2010*) and administered a single dose of tamoxifen at E14. Since approximately 6–12 hr typically elapse between tamoxifen administration and the onset of recombination (*Cai et al., 2013*; *Gridley and Groves, 2014*), this time point allows us to map the fate of Lfng-expressing cells in the apical and middle regions of the cochlea that we observe at the prosensory domain border. Mice were sacrificed four days later at E18.5, and sections and whole mounts of the cochlea were stained for the presence of hair cells, pillar cells and the EGFP reporter. In the most apical regions of the cochlea, we observed most EGFP-labeled cells in Kölliker's organ and very few in the inner hair cell region (*Figure 2A,B*; *Figure 2—figure supplement 1*), consistent with the expression of Lfng in this region between E12 and E14 (*Morsli et al., 1998*; *Ohyama et al., 2010*). In middle turn regions of the cochlea, we saw many EGFP-labeled inner hair cells and inner phalangeal cells (*Figure 2A,B*; *Figure 2—figure supplement 1*). In many cases, labeled inner hair cells were found adjacent to labeled inner phalangeal cells in whole mounts and sections. However, we observed very little labeling of other organ of Corti cell types in apical or middle turn regions (*Figure 2A,B*), suggesting that the strongly Lfng-expressing cells we observed at the border of the prosensory domain at E14-15 are progenitors of both inner hair cells and inner phalangeal cells. In basal regions of the cochlea,

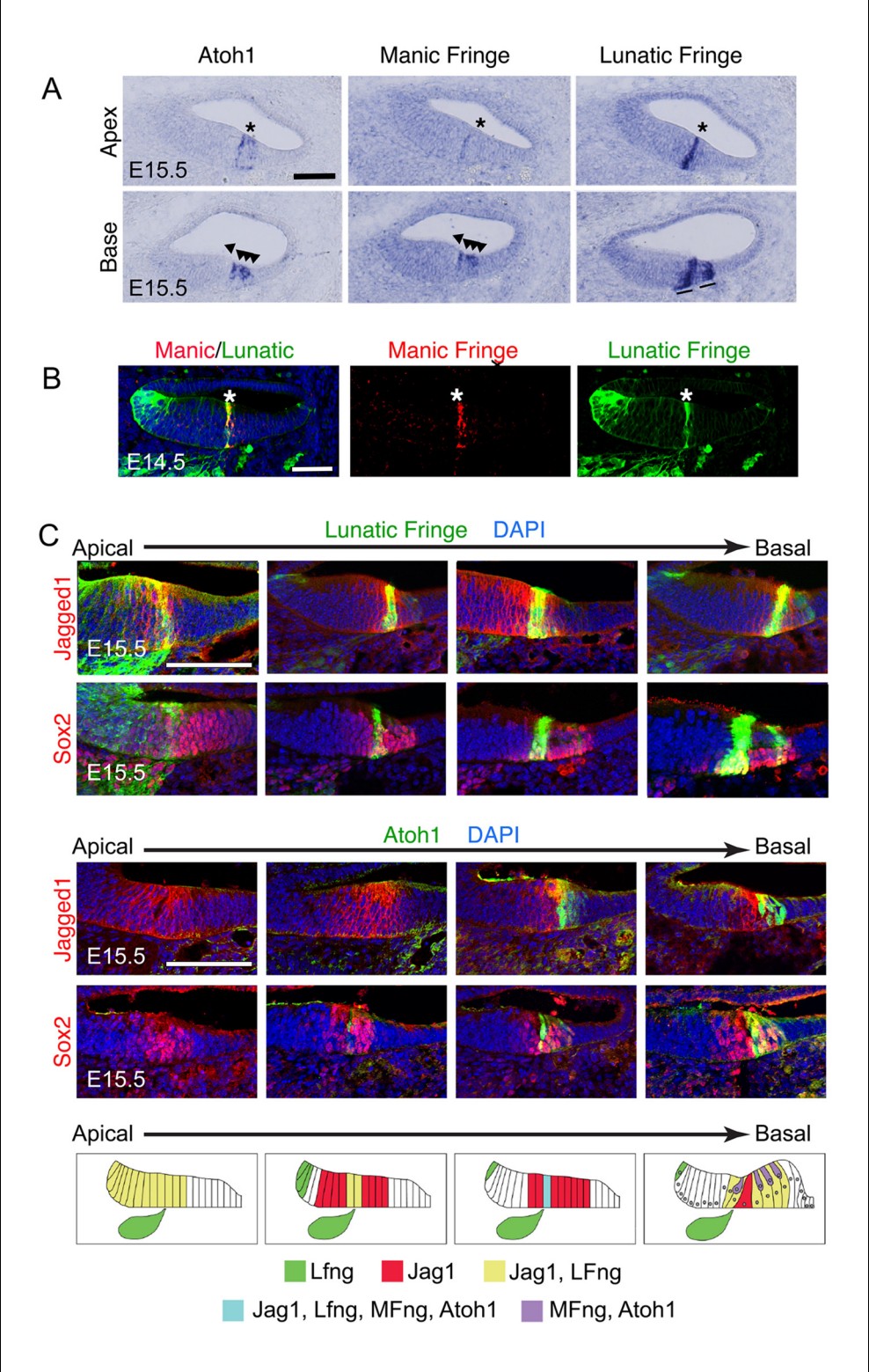

**Figure 1.** Dynamic expression of Lfng, Mfng and Jag1 during the onset of cochlear hair cell differentiation. (**A**) Serial sections from the basal and apical turns of an E15.5 cochlea processed for in situ hybridization for *Atoh1*, *Mfng* and *Lfng*. At the apex, where hair cell differentiation is just commencing, *Lfng* and *Mfng* co-localize to the first *Atoh1*-expressing cells at the boundary of the developing organ of Corti (asterisks). In the more mature basal region, *Mfng* is co-expressed with *Atoh1* in immature hair cells (arrowheads), while *Lfng* is restricted to supporting
*Figure 1 continued on next page*

*Figure 1 continued*

cells (black lines). A more complete series of sections is shown in *Figure 1—figure supplement 1* (B) *Mfng* and *Lfng* co-localize in the region of differentiating hair cells (asterisk). Fluorescent in situ hybridization for *Mfng* mRNA (red) was performed on cochlear sections from *Lfng-GFP* BAC transgenic mice, immunostained with antibodies to GFP (green). (C) Expression of Jag1 protein and *Lfng* in relationship to differentiating hair cells. Sections spanning the apical-basal axis of the E15.5 cochlea were taken from *Lfng-GFP* BAC transgenic mice and *Atoh1$^{GFP/GFP}$* knock-in mice. In each case, sections were stained with antibodies to GFP and to either Jag1 or Sox2 to mark prosensory cells and cells in Kölliker's organ. The dynamic expression pattern is summarized in (D) – Jag1 and Lfng are expressed in Kölliker's organ in the apex of the cochlea, then become restricted to supporting cells in the base. A stripe of Lfng and Mfng coincides with the first differentiating hair cells at the border of Kölliker's organ. After hair cell differentiation initiates, Atoh1 and Mfng are restricted to hair cells. The position of innervation from the Lfng-expressing spiral ganglion afferents (green ganglion) at the site of the first inner hair cells is indicated in each schematic panel.

The following figure supplement is available for figure 1:

**Figure supplement 1.** Detailed analysis of dynamic expression changes in *Lfng, Mfng* and *Atoh1* expression along the apical-basal axis of the E15.5 cochlear duct.

---

we now observed many labeled Deiters' cells, some labeled inner pillar cells as well as labeled inner hair cells and inner phalangeal cells (*Figure 2A*). These results are consistent with the onset of expression of Lfng in all supporting cell types except outer pillar cells in the basal region of the cochlea at E15.5 (*Figure 1A*). We also occasionally saw labeled outer hair cells (*Figure 2A,B*), suggesting that some Deiters' cell progenitors may differentiate into hair cells as Notch-mediated lateral inhibition establishes the precise pattern of hair cells and supporting cells in the organ of Corti. The transition from middle to basal turns of the cochlear duct was characterized by a gradual spreading of EGFP labeling into the Deiters' cell and outer hair cell region (*Figure 2A,B* and *Figure 2—figure supplement 1*).

Our data showing that *Lfng* and *Mfng* are transiently expressed together at the boundary of Kölliker's organ suggested they play a role in the differentiation of inner hair cells and their associated inner phalangeal cells at this boundary. We examined the cochleas of neonatal *Lfng;Mfng* double mutant mice and single *Lfng* or *Mfng* mutants. *Lfng* mutant mice have previously been shown to have no detectable cochlear phenotype (*Zhang et al., 2000*). *Mfng* mutant mice also had no cochlear phenotype (*Figure 2C*). In contrast, *Lfng;Mfng* double mutant mice showed significant numbers of supernumerary inner hair cells compared to controls (19.5 inner hair cells/100 μm compared to 12.3/100 μm in wild type controls; *Figure 2C*; *Figure 3B*). In contrast, we observed no significant changes in outer hair cell number (43.5 cells/100 μm versus 40.6 cells/100 μm in wild type controls; *Figure 2C*; *Figure 3C*). We were unable to detect expression of the third mammalian Fringe homologue, *Radical Fringe* (*Rfng*) in the cochlea, and *Lfng;Mfng;Rfng* triple homozygous mutants showed no significant difference in the number of supernumerary inner hair cells compared to *Lfng;Mfng* double mutant animals (*Figure 3B,C*).

The appearance of supernumerary hair cells is typically thought to occur by a failure of Notch-mediated lateral inhibition, whereby supporting cells trans-differentiate into hair cells, or their progenitors differentiate into hair cells instead of supporting cells (*Lewis 1998*; *Kiernan, 2013*). Since Fringe proteins act to modify Notch receptors and ligands to change their signaling properties (*Rana and Haltiwanger, 2011*), we examined the expression of supporting cell markers in our mutants. Surprisingly, we continued to observe p27$^{kip1}$-expressing supporting cells lying beneath the supernumerary inner hair cells, which is not predicted by a classical lateral inhibition model (*Figure 2C*). To identify the type of supporting cells beneath the inner hair cells, we used antibodies to the glutamate/aspartate transporter GLAST encoded by the *Slc1a3* gene and a nervous system-specific fatty acid binding protein, FABP7, both of which label inner phalangeal cells (*Furness and Lehre, 1997*; *Saino-Saito et al., 2010*; *Zilberstein et al., 2012*). We found both markers continued to be expressed in *Lfng;Mfng* mutants, and that supernumerary inner phalangeal cells were found adjacent to supernumerary inner hair cells (*Figure 2D*). Thus, in the absence of Fringe activity, both inner hair cells *and* their associated inner phalangeal cells were duplicated at the boundary of

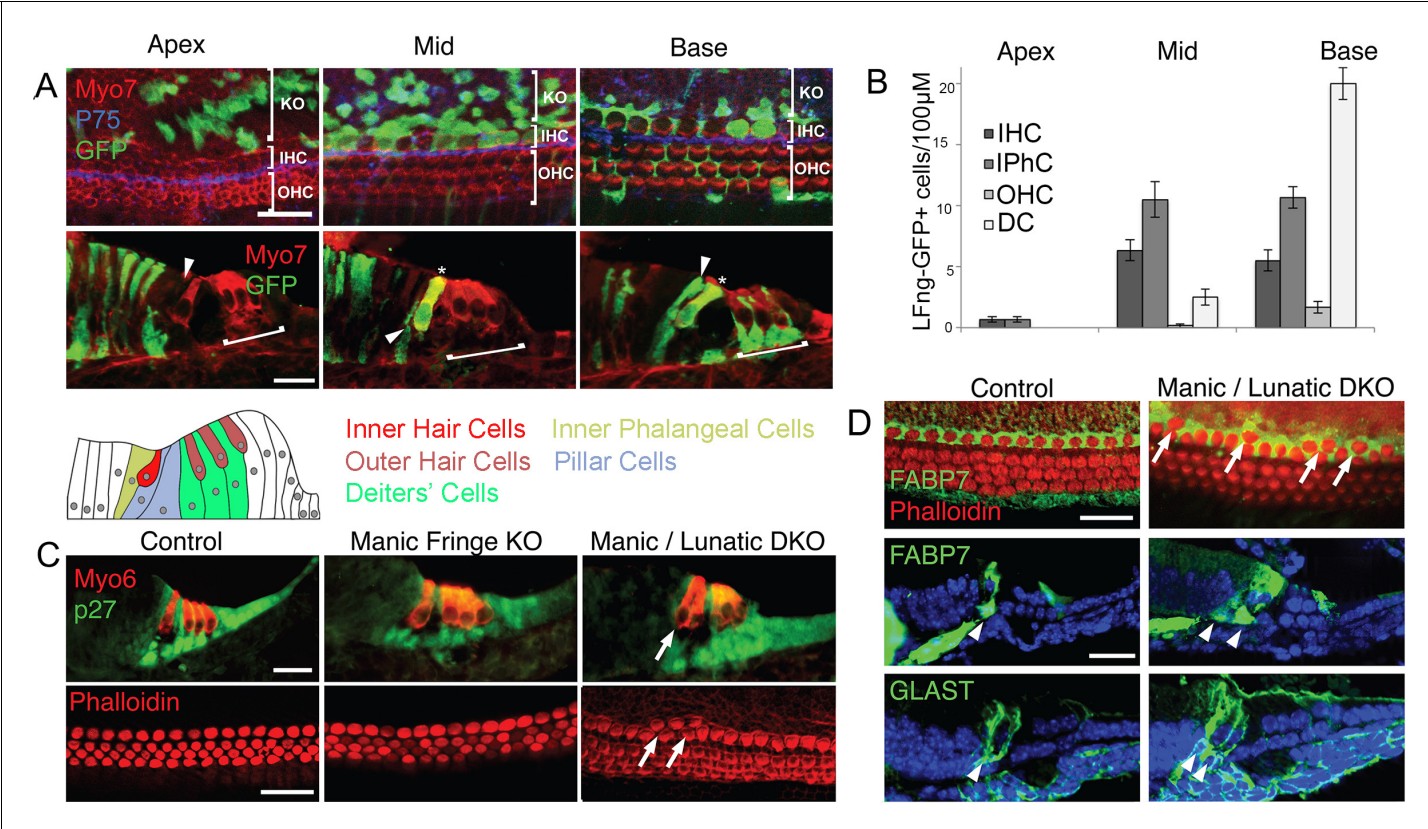

**Figure 2.** Fate mapping of Lfng progenitors and characterization of Lfng;Mfng mutant mice. (A) Fate mapping with Lfng-CreEr mice shows that inner hair cells and inner phalangeal cells derive from of Lfng-expressing progenitors. Lfng-CreER mice were crossed with Ai3 Cre reporter mice and tamoxifen was administered at E14.5. Mice were allowed to develop until E18.5 and then apical, mid-turn and basal regions of the cochlea examined in whole mount and sections. In the apical, most immature region of the cochlea, no organ of Corti cells are labeled; however robust labeling is seen in cells of Kölliker's organ. In the middle turn, both inner hair cells and their associated inner phalangeal cells are labeled, but no other cells in the organ of Corti are labeled. In the basal, most mature region of the cochlea, Deiters' cells and pillar cells are strongly labeled in addition to the inner hair cell region. Occasional outer hair cells are also labeled. (B) Quantitation of GFP-labeled cells in the organ of Corti along the apical-basal axis. In the apex, small numbers of inner hair cells (IHC) and inner phalangeal cells (IPhC) are labeled with GFP. In mid-turn regions, many more of these cells are labeled, together with a small number of outer hair cells (OHC) and Deiters' cells (DC). Deiters' cells constitute the majority of labeled cells in basal regions of the cochlea. (C): Duplication of inner hair cells and their associated supporting cells in *Lfng/Mfng* double mutants. Sections and whole mount preparations of P0 control, *Mfng*$^{-/-}$ and *Mfng*$^{-/-}$;*Lfng*$^{-/-}$ mutant mice. Sections show immunostaining for hair cells (Myosin6; red) and supporting cells (p27$^{kip1}$; green), while whole mount preparations reveal hair cell actin with fluorescently labeled phalloidin (red). *Mfng*$^{-/-}$;*Lfng*$^{-/-}$ mutant cochleas have regions containing ectopic inner hair cells (arrows). This phenotype is not observed in *Mfng*$^{-/-}$ embryos, (or *Lfng*$^{-/-}$ embryos; not shown, *Zhang et al., 2000*). (D) Duplication of inner hair cells in *Mfng*$^{-/-}$;*Lfng*$^{-/-}$ mutant cochleas is accompanied by a duplication of the surrounding inner phalangeal cells. Whole mount preparations of control and double mutant cochleas are stained with fluorescently-labeled phalloidin (red) and antibodies to FABP7 (green) to label inner phalangeal cells. Sections of control and double mutant cochleas are stained with antibodies to either FABP7 or GLAST to reveal inner phalangeal cells. The duplicated inner phalangeal cell region is indicated with arrows.

The following figure supplement is available for figure 2:

**Figure supplement 1.** Lineage tracing with *LFng-CreER* transgenic mice recapitulates the dynamic pattern of *Lfng* expression.

Kölliker's organ and the prosensory domain. This suggests that the defects we observe in these mutant mice do not result from a failure of Notch-mediated lateral inhibition that has been previously demonstrated to distinguish between hair cells and supporting cells (*Kiernan, 2013*). Instead, it appears that the column of inner hair cell and inner phalangeal cell progenitors that transiently express *Lfng* and *Mfng* normally inhibit their neighbors in Kölliker's organ from also adopting an inner hair cell/inner phalangeal cell fate.

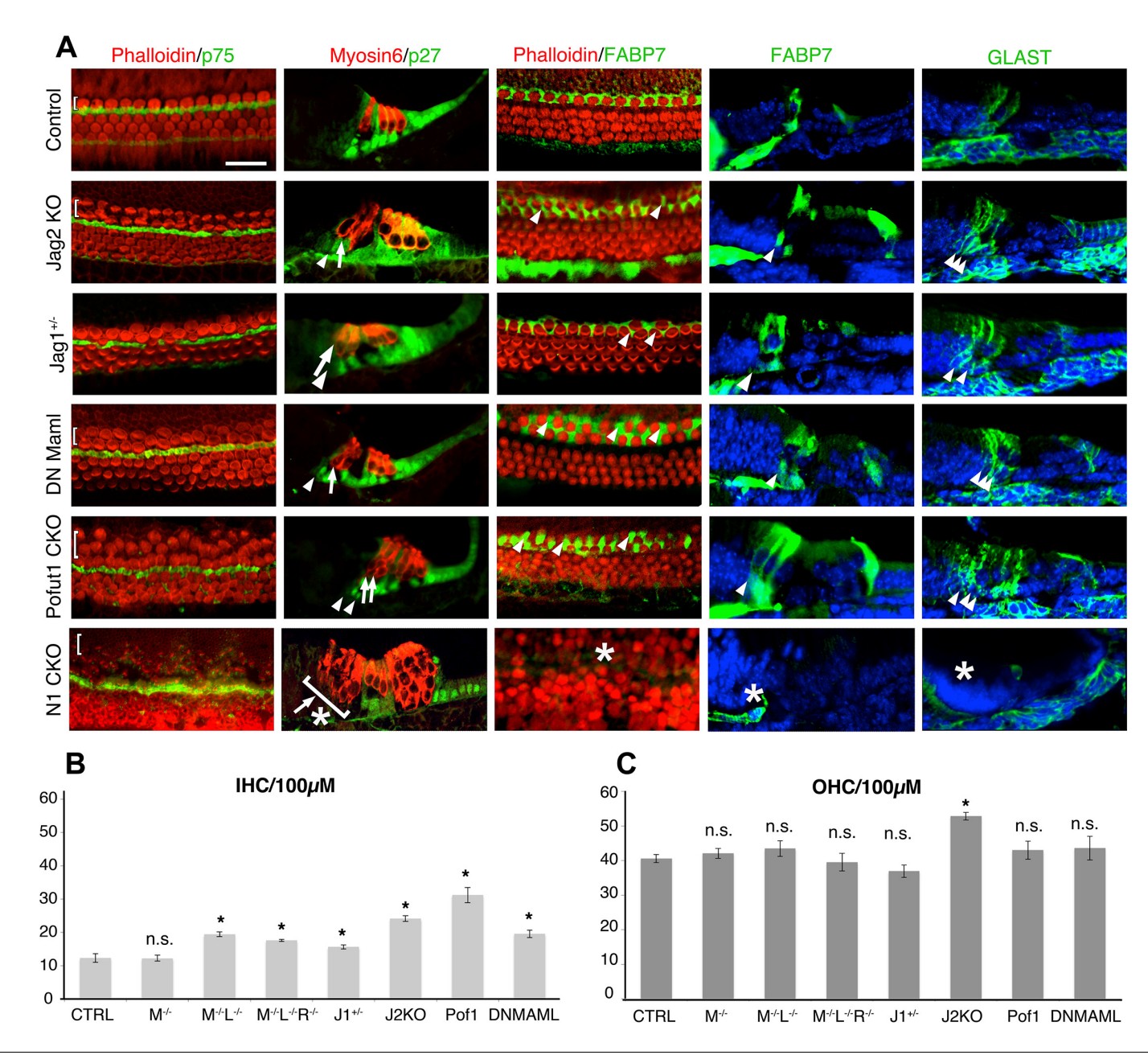

**Figure 3.** Duplication of inner hair cells and inner phalangeal cells in hypomorphic Notch loss-of-function alleles. (**A**) *Jag2*$^{-/-}$ mutants; *Jag1*$^{-/+}$ heterozygotes and conditional *Pofut1* and *dnMAML1* mutants all display regions of duplicated inner hair cells and inner phalangeal cells. The inner hair cell region is shown on P0 cochlear whole mount preparations stained with phalloidin (hair cells; red) and either p75 (pillar cells; green) or FABP7 (inner phalangeal cells; green). Sections show all hair cells and supporting cells (Myosin6; red and p27$^{kip1}$; green) or just inner phalangeal cells (FABP7 or GLAST; green). Ectopic inner phalangeal cells are shown with white arrows. White asterisks show absent supporting cells. (**B**) Inner hair cell numbers are significantly increased in *Mfng*$^{-/-}$;*Lfng*$^{-/-}$ mutants, *Mfng*$^{-/-}$;*Lfng*$^{-/-}$; *Rfng*$^{-/-}$ mutants, *Pofut1* and *dnMAML1* conditional mutants and *Jag1*$^{+/-}$ and *Jag2*$^{-/-}$ mutants, but not *Mfng*$^{-/-}$ or control cochleas. (**C**) Outer hair cells numbers are only significantly increased in *Jag2*$^{-/-}$ mutants. In each case, bars represent the mean number of inner or outer hair cells per 100 µm (p<0.05; Student two-tailed t test). M: *Mfng;* L: *Lfng,* R: *Rfng;* Pof1: *Pofut1;* DNMAML: *dnMAML1;* J1: Jag1; J2: *Jag2.*

The following figure supplement is available for figure 3:

**Figure supplement 1.** Additional characterization of Pofut1, dnMAML1 and GLAST-DsRed mice.

## Reduction in Notch signaling leads to the formation of supernumerary inner hair cells and inner phalangeal cells

Modification of Notch receptors by Fringe proteins can cause changes in the strength of Notch signaling (*Haltiwanger and Stanley, 2002*). Differentiating hair cells express three Notch ligands, Dll1, Dll3 and Jag2 (*Lanford et al., 1999*; *Morrison et al., 1999*; *Hartman et al., 2007*; *Maass et al., 2015*), of which Dll1 and Jag2 act as conventional Notch ligands, whereas Dll3 appears to act only by inhibiting Notch receptors in the same cell (*Ladi et al., 2005*; *Geffers et al., 2007*; *Chapman et al., 2011*). To determine whether the loss of Fringe proteins led to a reduction in Notch signaling delivered by differentiating hair cells, we compared the *Lfng;Mfng* phenotype to the cochlear phenotype seen in *Jag2* mutants. *Jag2* mutant mice display supernumerary inner hair cells (*Lanford et al., 1999*; *Zhang et al., 2000*; *Figure 3A,B*), and we also observed supernumerary inner phalangeal cells in these mice on the basis of p27$^{kip1}$, GLAST and FABP7 immunostaining (*Figure 3A*). The Jag1 ligand is also expressed in the cochlear duct as the first hair cells differentiate (*Figure 1C*), and later in supporting cells (*Morrison et al., 1999*; *Woods et al., 2004*). *Jag1* heterozygous mice also have increased inner hair cells (*Kiernan et al., 2007*; *Moayedi et al., 2014*), and we observed a similar phenotype of duplication of inner hair cells and inner phalangeal cells in *Jag1* heterozygous mice (*Figure 3A,B*). We also confirmed that loss of both *Lfng* and *Mfng* led to a reduction in Notch signaling directly, by comparing N1ICD staining in *Lfng;Mfng* embryos to wild type controls (*Figure 4A*). In contrast to other regions of *Lfng;Mfng* mutant embryos such as the hindbrain (*Figure 4A*, inset), *Lfng;Mfng* mutants showed a reduction in N1ICD staining in the E15.5 cochlear duct.

We next asked whether the observed inner hair cell and inner phalangeal cell duplication phenotype could be recapitulated by a more general reduction in Notch signaling. To do this, we

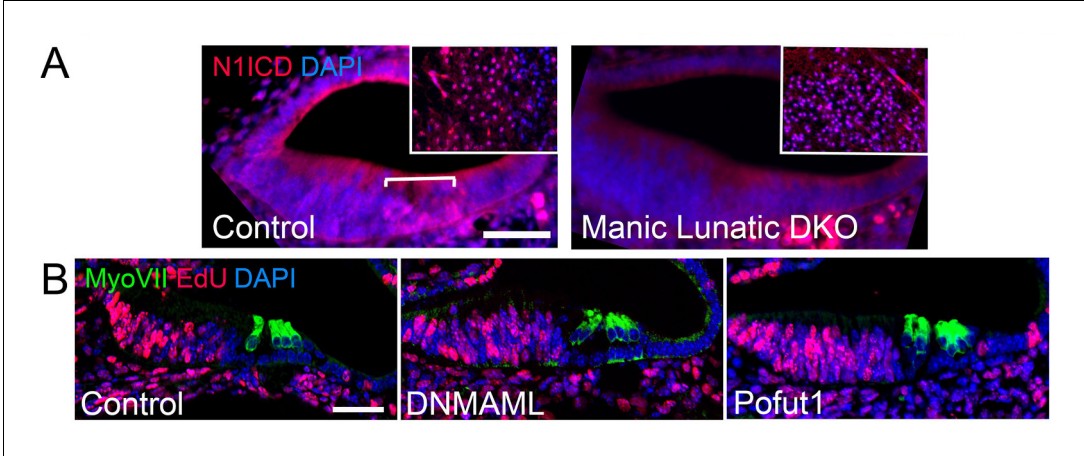

**Figure 4.** Notch signaling and EdU incorporation in Notch loss-of-function mutants. (**A**) Notch signaling is reduced in *Lfng;Mfng* double mutants. Sections through the cochlear duct of E15.5 wild type and *Lfng;Mfng* double mutant are shown stained for the Notch1 intracellular domain (N1ICD). The insets show N1ICD staining in the hindbrain taken from the same section containing the cochlear duct as a positive control. In *Lfng;Mfng* double mutants, N1ICD staining is greatly reduced in the cochlea, but not the hindbrain. (**B**) Edu incorporation shows no significant increase in labeling of inner hair cells or inner phalangeal cells following a reduction in Notch signaling. We administered EdU to pregnant female mice three times a day between E14.5 and E17.5 and collected embryos for analysis at E18.5. We observed no significant increase in EdU incorporation in the inner hair cells or inner phalangeal cells in *Pofut1* and *dnMAML1* mutant embryos compared to wild type controls. We occasionally saw EdU incorporation in border cells in *dnMAML1* mutant embryos (asterisk). Data for EdU incorporation is provided in *Table 1*.

The following figure supplement is available for figure 4:

**Figure supplement 1.** Supernumerary hair cells are present from the onset of hair cell differentiation in mutants that reduce Notch signaling.

examined two mouse mutants that displayed a partial loss of Notch signaling in the cochlea. Pofut1 is an O-fucosyltransferase that adds fucose moieties to Notch receptors and ligands (*Takeuchi and Haltiwanger, 2010*), and whose activity generates the O-fucose substrate that is necessary for further N-glycosylation of Notch receptors by Fringe proteins (*Shi and Stanley, 2003*; *Shi et al., 2005*). *Pofut1* null mutant mice die between E9.5 and E10.5 with an axial phenotype reminiscent of a Notch loss of function (*Shi and Stanley, 2003*), although different tissues can display differential sensitivities to loss of *Pofut1* (*Schuster-Gossler et al., 2009*). Conditional deletion of *Pofut1* in the cochlea with *Pax2-Cre* transgenic mice leads to a reduction, but not a complete loss of Notch signaling, as revealed by the presence of cleaved Notch1-intracellular protein in the nuclei of *Pofut1* mutant supporting cells and the enhancement of the *Pofut1* mutant phenotype by additional Notch inhibition with gamma secretase inhibitors in organ culture (*Figure 3—figure supplement 1A,B*). This incomplete loss of Notch signaling suggests that at least some Notch receptors lacking O-fucose glycans can still be chaperoned to the surface of progenitor cells in the *Pofut1* mutant cochlea. *Pofut1* conditional mutant mice also exhibited both supernumerary inner hair cells and inner phalangeal cells (*Figure 3A,B*), suggesting that this phenotype can be caused by a general reduction in Notch signaling. To confirm this, we also examined transgenic mice carrying a Cre-inducible dominant-negative form of MAML1, a nuclear co-factor responsible for the formation of transcriptional complexes between the Notch intracellular domain and the RBPJ co-factor (*Kovall, 2008*). Embryos derived by mating *Pax2-Cre* and Cre-inducible *dnMAML1* mice (*Tu et al., 2005*) also showed only a partial Notch loss of function (*Figure 3—figure supplement 1A,B*) and displayed both supernumerary inner hair cells and inner phalangeal cells (*Figure 3A*). We found that *Lfng;Mfng* mutants, *Jag2* mutants, *Pofut1* conditional mutants and *dMAML1* mutants all showed a significant increase in inner hair cells and the presence of extra inner phalangeal cells (*Figure 3A,C*), but we only observed an increase in outer hair cells in *Jag2* mutants (*Figure 3B*; *Lanford et al., 1999*).

Our results suggested that the level of Notch signaling required to restrict the number of inner hair cells and inner phalangeal cells to a single row during cochlear development is very sensitive to manipulation, such that even a partial reduction causes duplication of the inner hair cell/inner phalangeal cell region. However, the persistence of inner phalangeal cells in our mutant lines (*Figure 2B*; *3A*) suggest that Notch-mediated lateral inhibition between hair cells and supporting cells still occurs in our various mouse models when Notch signaling is reduced but not abolished. To test this, we examined *Notch1* conditional null mutant mice which have been shown to produce a significant loss of lateral inhibition between hair cells and supporting cells in the cochlea (*Kiernan et al., 2005*). As expected, we observed a very large increase in both inner and outer hair cells, and a great reduction of many kinds of supporting cells (*Figure 3A*). Significantly, we saw no evidence for inner phalangeal cells in Notch1 conditional mutants on the basis of p27$^{kip1}$, GLAST or FABP7 staining (*Figure 3A*), suggesting that the cochlear phenotype caused by a strong Notch loss of function is due to a combination of an increase in inner hair / inner phalangeal progenitor specification and the subsequent differentiation of these ectopic progenitors exclusively into inner hair cells.

The supernumerary inner hair cells and inner phalangeal cells observed when Notch function is reduced could arise by a change in cell fate within the cochlear duct, or alternatively by the prolonged proliferation of progenitors at the border of the prosensory domain and Kölliker's organ. Indeed, previous studies have observed a small amount of prolonged proliferation of cochlear progenitors in Notch pathway mutants (*Kiernan et al., 2005*; *Murata et al., 2009*; *Tateya et al., 2011*). To distinguish between these possibilities, we administered EdU to pregnant female mice three times a day between E14.5 and E17.5 and collected embryos for analysis at E18.5. We performed these experiments in wild type, *Pofut1* and *dnMAML1* conditional mutant mice (*Figure 4*). We did not observe any significant differences in EdU labeling of outer hair cells, Deiters' cells, pillar cells, inner hair cells or inner phalangeal cells (*Table 1*). We observed a modest but significant increase in EdU incorporation in *dnMAML1* border cells immediately adjacent to inner phalangeal cells (66 versus 59 labeled cells; p=0.014). To test at what stage the supernumerary inner hair cells and inner phalangeal cells arose in Notch pathway mutants, we examined *Atoh1* expression at the leading edge of hair cell differentiation in the apex of wild type and *Pofut1* mutant cochleas at E15.5. We consistently observed doublets of *Atoh1*-expressing cells in this region of *Pofut1* mutants, whereas single *Atoh1*-expressing cells were always observed at the leading edge of hair cell differentiation in wild type cochleas (*Figure 4—figure supplement 1*). These results suggest that the supernumerary

**Table 1.** EdU labeling of cochlear progenitor cells in two Notch loss-of-function mutants.

| | Number of cochleas | Number of sections | Total EdU labeled cell types | | | | | |
| --- | --- | --- | --- | --- | --- | --- | --- | --- |
| | | | IHC | OHC | BC | IPC | PC | DC |
| Control | 11 | 314 | 0 (0) | 3 (0.009) | 59 (0.187) | 11 (0.035) | 2 (0.006) | 9 (0.028) |
| *dnMAML*1 Mutant | 9 | 239 | 1 (0.004) | 5 (0.02) | 66* (0.276) | 10 (0.041) | 2 (0.008) | 5 (0.020) |
| *Pofut1* Mutant | 4 | 103 | 0 (0) | 1 (0.019) | 24 (0.233) | 8 (0.077) | 0 (0) | 4 (0.038) |

We administered EdU to pregnant female mice three times a day between E14.5 and E17.5 and collected embryos for analysis at E18.5. The total numbers of dividing cells labeled by EdU for each genotype was normalized by dividing the number of labeled cells by the total number of sections counted. The total number counted for all sections is shown under each cell type and the normalized number per section is shown below in parentheses. A modified Wald test for two-sample proportions was used to determine whether the numbers of dividing cells was significantly different in either mutant group compared with the control groups. Statistical tests were applied to individual hair and supporting cell types (see text). The only group that showed significant differences to control was the number of labeled border cells in *dnMAML1* mutants (*p=0.014). IHC: Inner hair cells; OHC: Outer hair cells; BC: Border cells; IPC: inner phalangeal cells; PC: Pillar cells; DC: Deiters' cells.

inner hair cells are induced at the same time as their normal counterparts when Notch signaling is reduced in the cochlea.

To further demonstrate that the restriction of inner hair cell and inner phalangeal cell numbers was more sensitive to changes in Notch signaling than that required for the lateral inhibition of supporting cell fate by hair cells, we established an in vitro cochlear culture system in which we attenuated Notch signaling to different degrees. We used double transgenic mice carrying a *GLAST-dsRed* transgene (*Regan et al., 2007*) to label inner phalangeal cells (*Figure 3—figure supplement 1C*) and *Atoh1-GFP* reporter mice (*Shroyer et al., 2007*; *Cai et al., 2013*, *2015*). We isolated cochleas from E14.5 double transgenic mice and cultured them for three days in the presence of different doses of the gamma secretase inhibitor DAPT or Notch1 blocking antibodies (*Wu et al., 2010*; *Maass et al., 2015*). We demonstrated the differential degree of Notch inhibition by measuring levels of Notch-responsive *Hes* and *Hey* genes in our cultures. We observed a modest down-regulation of *Hes1* and *Hes5* at intermediate doses of DAPT or anti-Notch1 antibodies, but no significant down-regulation of *Hey1*, *Hey2* or *Heyl* (*Figure 5B,E*). However, at high concentrations of either inhibitor, all genes were significantly down-regulated with the exception of *Hey2*, which is much less sensitive to changes in Notch signaling in the cochlea (*Doetzlhofer et al., 2009*). At high concentrations of either inhibitor, we observed significant increases in both outer and inner hair cells, and a loss of inner phalangeal cells as revealed by the absence of GLAST-dsRed fluorescence (*Figure 5A, D*). This result is consistent with a loss of lateral inhibition between hair cells and supporting cells (*Kiernan, 2013*). In contrast, when we applied 100-fold lower concentrations of Notch1 antibody or 25-fold lower doses of DAPT to the cultures, we saw a smaller but still significant increase in the number of inner hair cells, but no significant increase in outer hair cell numbers (*Figure 5C,F*). Moreover, we continued to observe expression of the *GLAST-dsRed* transgene at these doses (*Figure 5A,D*), suggesting that inner phalangeal cells do not trans-differentiate into hair cells, but remain adjacent to the supernumerary inner hair cells.

## Discussion

The Notch signaling pathway acts at multiple stages to regulate the development of the inner ear. It was first proposed to regulate the production of cochleo-vestibular ganglion neurons, and later of hair cells and supporting cells by lateral inhibition (*Lewis, 1991*; *Haddon et al., 1998*; *Lewis 1998*). Notch signaling can also regulate the production of sensory patches though inductive signaling between Jag1 and Notch, leading to the up-regulation of Sox2 (*Eddison et al., 2000*; *Hartman et al., 2010*; *Pan et al., 2010*; *Neves et al., 2011*, *2013*; *Pan et al., 2013*). In this study, we identify a new and unexpected role for Notch signaling in the ear: the positioning of inner hair cells and their associated inner phalangeal cells at the boundary of Kölliker's organ and the

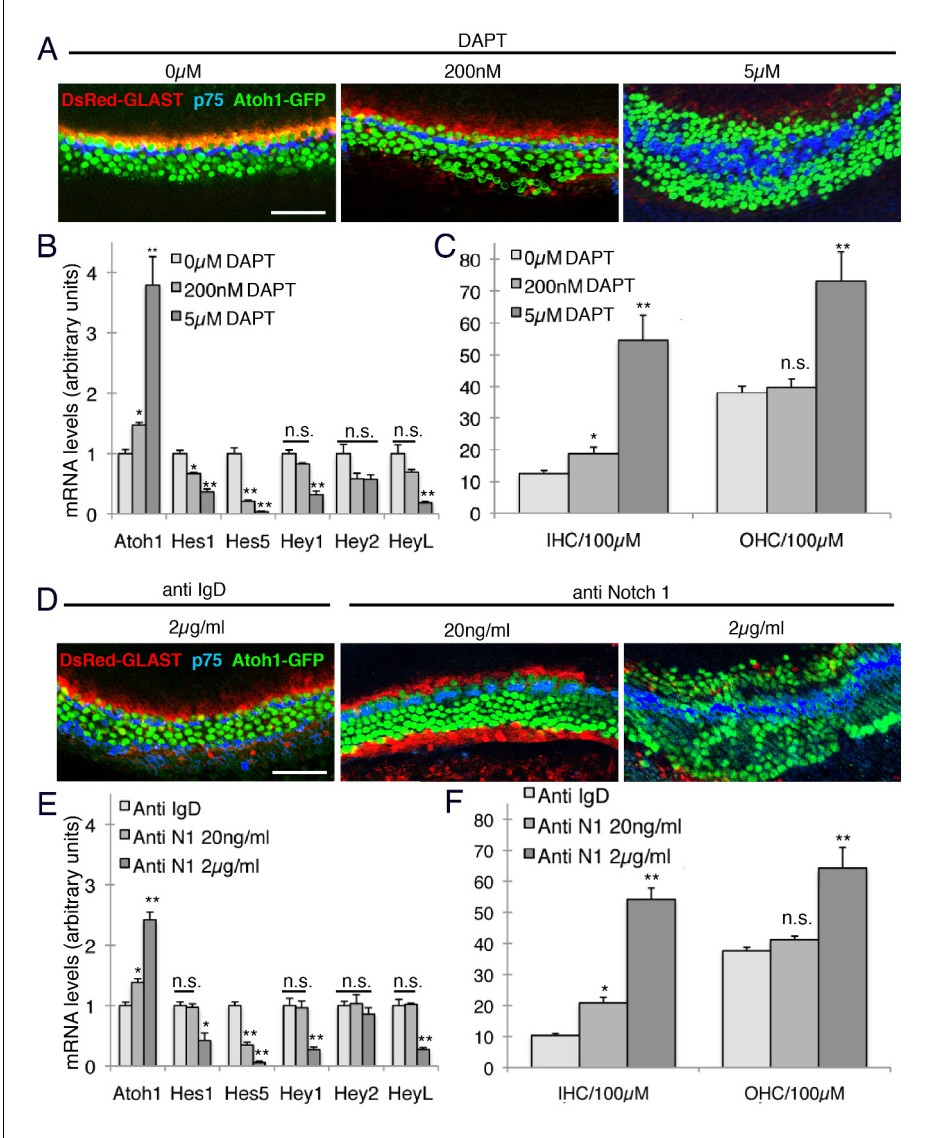

**Figure 5.** Intermediate doses of Notch inhibitors cause ectopic inner hair cells but a persistence of inner phalangeal cells. E14 cochleas from *GLAST-DsRed;Atoh1*<sup></sup>*GFP/GFP* mice were cultured for 24 hr in the presence of different doses of either (**A**) DAPT (0, 200 nM or 5 μM) or (**B**) Notch1 blocking antibodies (0, 20 ngml or 2 μg/ml). Inner and outer hair cells were quantified from the explants, with p75$^{LNGFR}$ antibody staining (blue) used to reveal pillar cells, and parallel cultures were taken for quantification of *Atoh1* mRNA and the Hes/Hey genes *Hes1, Hes5, Hey1, Hey2* and *Heyl*. For both DAPT and Notch1 antibodies, intermediate doses caused a significant increase in inner hair cell numbers but not outer hair cells, and a persistence of DsRed-expressing inner phalangeal cells (p<0.05; Student two-tailed t test).

prosensory domain. We show that this developmental decision is exquisitely sensitive to changes in Notch signaling. We observed a striking duplication of both inner hair cells *and* inner phalangeal cells in five different mutant mouse models and two organ culture systems that lower, but do not eliminate, Notch signaling in the cochlea. Unlike previous studies of Notch signaling in the cochlea, these phenotypes do not resemble defects in Notch-mediated lateral inhibition that is known to occur between hair cells and supporting cells (*Kiernan, 2013*). Only in conditions where Notch signaling is greatly reduced - such as in *Notch1* mutants or at high doses of Notch blocking antibodies or gamma secretase inhibitors – do we see a conversion of duplicated inner phalangeal cells into inner hair cells, suggestive of a failure of lateral inhibition between hair cells and supporting cells.

Our data, together with the careful analysis of Notch1 activation in the developing cochlea performed by Murata and colleagues (*Murata et al., 2006*) lead us to propose a new model for the development of the organ of Corti that involves two phases of Notch signaling. Because hair cell differentiation proceeds in a basal-apical direction over a period of several days (*Chen et al., 2002*; *Lee et al., 2006*; *Cai et al., 2013*), we discuss the timing of these events with respect to the basal region of the cochlea (*Figure 6*).

During the outgrowth of the cochlear duct, *Jag1* and *Lfng* are expressed in cochlear progenitor cells (*Morsli et al., 1998*; *Ohyama et al., 2010*). Lfng activity is known to attenuate Jag1-Notch signaling between cells (*Hicks et al., 2000*; *Rana and Haltiwanger, 2011*; *LeBon et al., 2014*) which explains why these cochlear progenitor cells have been reported to receive low amounts of Notch signal (*Figure 6A*; *Murata et al., 2006*). We propose that starting at approximately E13, the cochlear duct receives hair cell-inducing signals that reach a peak spanning the boundary of Kölliker's organ and the prosensory domain (*Figure 6B*). There is no consensus as to the identity of these signals; they may be positively acting signals such as Wnts (*Shi et al., 2010*; *Jacques et al., 2012*; *Shi et al., 2014*; *Jansson et al., 2015*) or may reflect the withdrawal of inhibitors of hair cell induction, such as Shh, which is expressed in the neurons invading the cochlear duct in this region and is cleared from the neurons in the same basal-apical gradient as hair cell differentiation (*Bok et al., 2013*; *Tateya et al., 2013*). In response to these hair cell-inducing signals, a column of prosensory cells adjacent to Kölliker's organ begins to up-regulate early hair cell genes (*Atoh1, Mfng, Jag2, Dll1*), while still maintaining expression of *Notch1, Lfng* and *Jag1* (*Figure 6B*, green cell). This single column of differentiating hair cell progenitors sends Notch signals to its neighbors in Kölliker's organ that prevent them from responding to hair cell-inducing signals (*Figure 6B*). Our Lfng-CreER fate mapping data show that this column of cells will ultimately form inner hair cells and their neighboring inner phalangeal cells (*Figure 2A,B*).

We suggest this interaction sets the boundary of the organ of Corti and establishes a single row of inner hair cells and inner phalangeal cells. In our study, we experimentally perturbed this interaction by either reducing the amount of Notch signaling *received* by Kölliker's organ cells (*Pofut1* mutants or *dnMAML1* mutants) or by reducing the amount of Notch signaling *delivered* by the adjacent column of prosensory cells (*Jag1* and *Jag2* mutants). Our observation that the duplication of this column is caused by reduction, but not elimination of Notch signaling in several mouse mutants and in cell culture (*Figures 3* and *5*) suggests this phase of Notch signaling involves *moderate or low* levels of Notch activity. We explain the effects of these perturbations on our model below.

*Pofut1* mutants or *dnMAML1* mutants cause a general reduction in Notch signaling received by the Kölliker's organ cells. As a result, they can now respond to hair cell inducing signals, leading to the formation of a second column of cells that express hair cell genes (*Atoh1, Mfng, Jag2, Dll1*) and maintain expression of *Notch1, Lfng* and *Jag1*. This second column of cells ultimately differentiates into a second row of inner hair cells and inner phalangeal cells. Mutating *Jag1* or *Jag2* reduces the available pool of Notch ligands and hence reduces the amount of Notch signaling delivered to Kölliker's organ, also causing a duplication phenotype. These results are summarized in *Figure 7*. Although we did not examine *Dll1* conditional mutants in our study, the co-expression of *Dll1* with *Jag2* in hair cell progenitors, together with previous reports of Dll1 loss of function in the cochlea (*Kiernan et al., 2005*; *Brooker et al., 2006*) suggest these mutants will also have a duplicated row of inner hair cells and inner phalangeal cells.

Fringe proteins are known to regulate Notch signaling by making Notch receptors more sensitive to Delta-type ligands and less sensitive to Jagged/Serrate-type ligands (*Haltiwanger and Stanley, 2002*). We show that *Lfng* has a very dynamic expression pattern (*Figure 1* and *Figure 1—figure supplement 1*) during cochlear development, but that it coincides with *Mfng* and *Atoh1* precisely where inner hair cells and inner phalangeal cells differentiate at the boundary of the organ of Corti. We confirmed this observation by lineage tracing with *Lfng-CreER* mice (*Figure 2A,B*, *Figure 2—figure supplement 1*). *Lfng;Mfng* double mutants show a reduction in Notch signaling at this stage of overlap and display a duplication of inner hair cells and inner phalangeal cells (*Figure 2C,D*). These results are consistent with the column of progenitor cells transiently expressing *Lfng, Mfng, Notch1, Atoh1, Dll1, Jag1* and *Jag2* sending less Notch signal to their neighbors in Kölliker's organ in *Lfng;Mfng* double mutants. How can we explain this result?

Previous studies suggest that when Notch ligands and receptors are present in the same cell, they attenuate each other's activity, a phenomenon known as *cis*-inhibition (*del Alamo and*

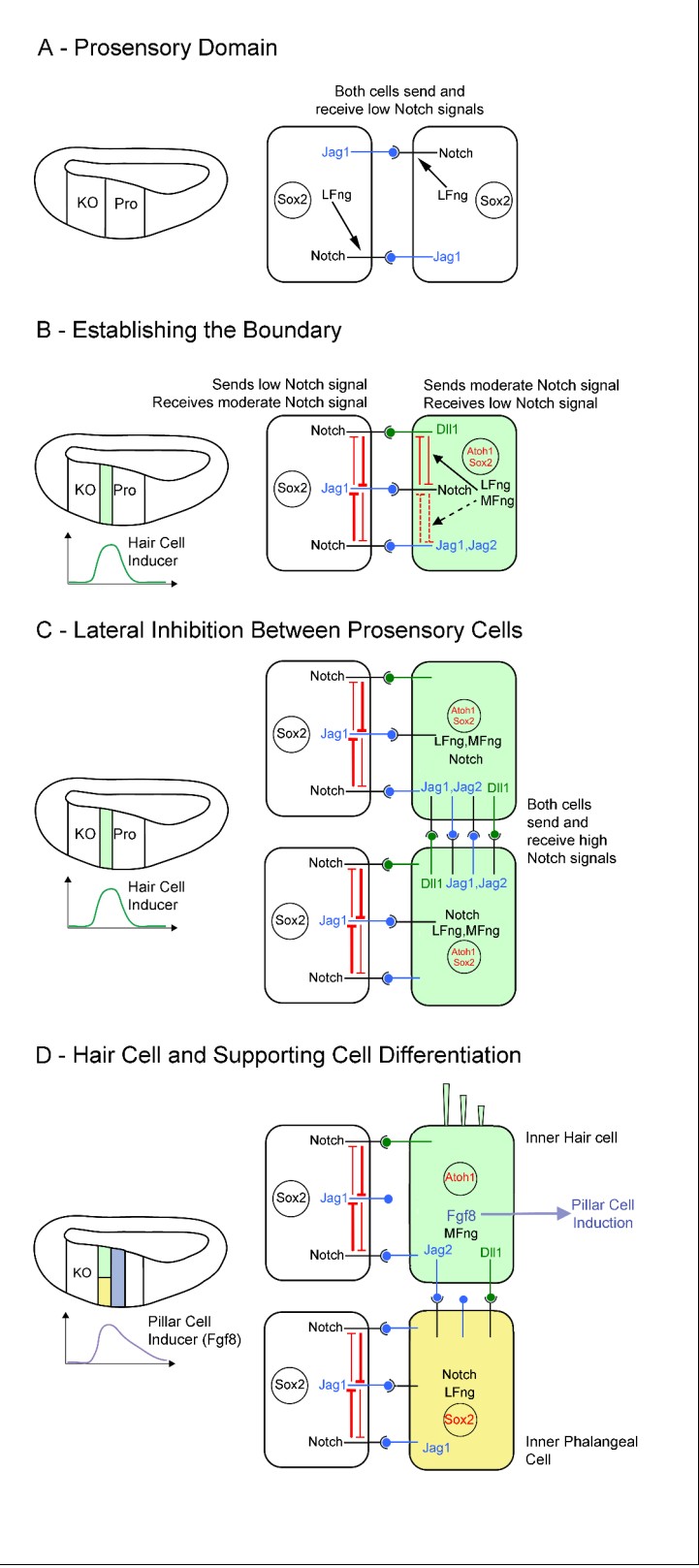

**Figure 6.** Model of Notch function during boundary formation between the prosensory domain and Kölliker's organ. (**A**): Between E11 and E13, cells in the cochlear primordium express both Lfng and Jag1 in the presumptive Kölliker's organ (KO) and prosensory domain (Pro). Lfng-mediated attenuation of Jag1-Notch1 signaling in trans leads to very low levels of Notch signaling in these cells. (**B**): Starting in the base of the cochlea

*Figure 6 continued*

at E13.5, hair cell inducing signals peak at the boundary of Kölliker's organ and the prosensory domain, leading to the up-regulation of *Mfng, Jag2, Dll1, Dll3* and *Atoh1* in a column of cells at the boundary (light green). We predict that the co-expression of *Lfng* and *Mfng* in these cells modulates the activity of Dll1 and Jag ligands in these cells through *cis*-inhibition (black arrows), The expression of *Lfng* and *Mfng* in these cells also makes them less sensitive to Jag1 signaling from neighboring cells in Kölliker's organ (white cell). (**C**): As a column of hair cell progenitors is differentiating at the prosensory-Kölliker's organ boundary (light green cells), lateral inhibition *within* this column of cells is carried out by Dll and Jag ligands. This typical lateral inhibition leads to (**D**) segregation into an inner hair cell (light green) and a supporting cell (yellow). The presence of Lfng and Mfng in this column leads to strong signaling from Dll1, and weak signaling from Jag1 and Jag2. The differentiating inner hair cells begin to express Fgf8, which induces neighboring cells to adopt a pillar cell fate (purple).

*Schweisguth, 2009*; *Becam et al., 2010*; *del Álamo et al., 2011*; *LeBon et al., 2014*). Fringe proteins have been shown to modulate *cis*-inhibition: in vertebrates, Lfng and Mfng proteins *increase cis*-inhibition between Notch and Delta ligands, but *decrease cis*-inhibition between Notch and Jagged ligands (*LeBon et al., 2014*). We therefore predict that the prosensory cell column adjacent to

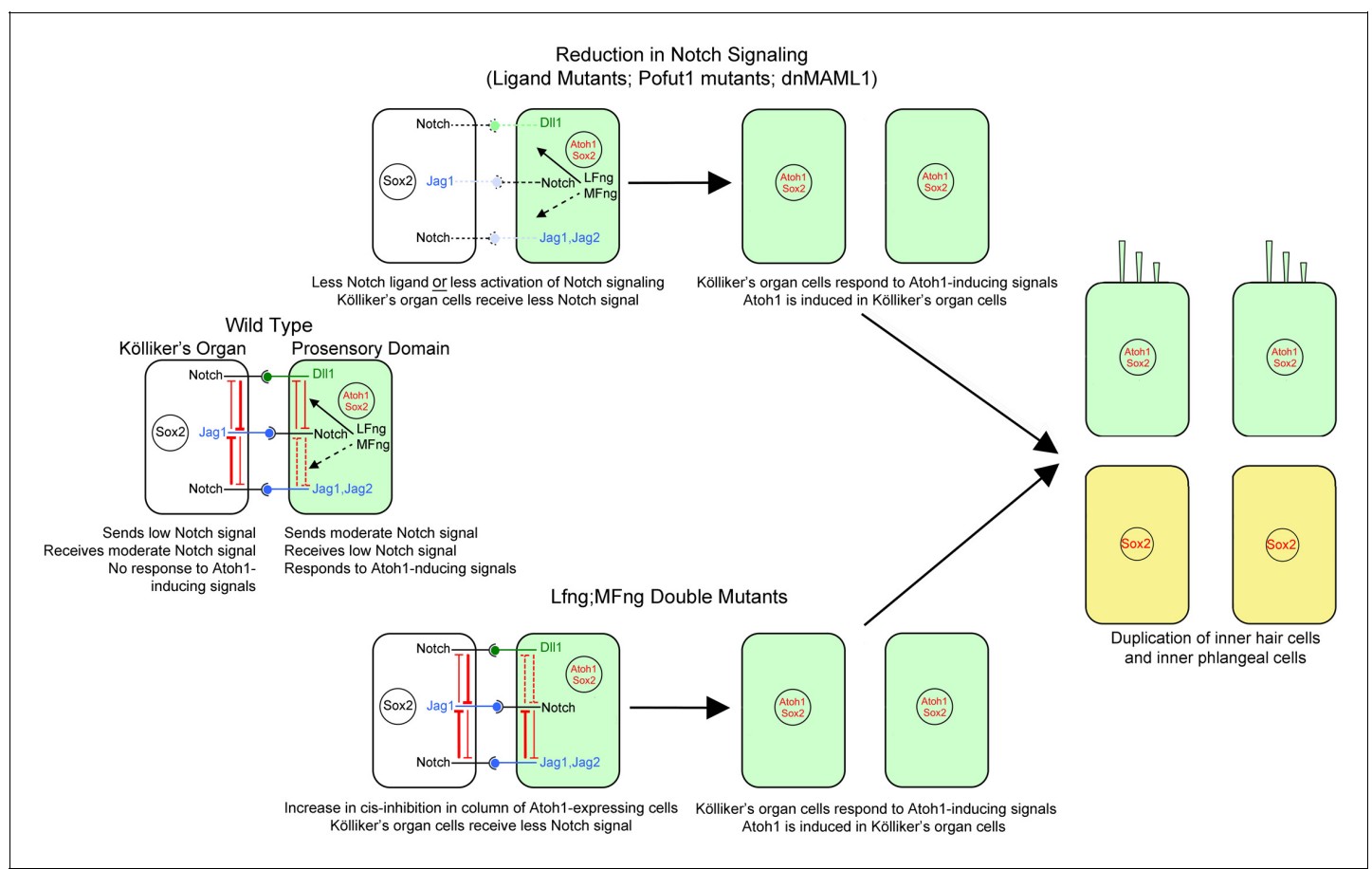

**Figure 7.** Mechanism of the Notch pathway mutants in this study. The wild type situation is shown on the left, where prosensory cells that receive hair cell inducing signals (green) deliver a moderate Notch signal to Kölliker's organ cells (white) and prevent them from adopting a prosensory or hair cell fate. In Notch ligand mutants (Jag1, Jag2), prosensory cells deliver less Notch signal to Kölliker's organ. In mutants that affect the Notch receptor (Pofut1) or intracellular Notch signaling (dnMAML1), less signal is received by Kölliker's organ. In all cases (center, top), one row of Kölliker's organ cells now responds to Atoh1 inducing signals and duplicate the inner hair cell and inner phalangeal cell (right image). In cells lacking *Lfng* and *Mfng*, (center, bottom), increased *cis*-inhibition in the prosensory region is predicted to deliver less Notch signal to Kölliker's organ. Once again, these cells respond to Atoh1 inducing signals and duplicate the inner hair cell and inner phalangeal cell (right image).

Kölliker's organ that expresses *Jag1, Jag2, Dll1, Notch1, and Lfng* and *Mfng* experiences *cis*-inhibition (*Figure 6B*, red inhibitory lines), leading to only a moderate amount of Notch signal delivered to Kölliker's organ. Loss of *Lfng* and *Mfng* is predicted to cause more cis-inhibition between Notch1 and Jag1 and Jag2, and less *cis*-inhibition between Notch1 and Dll1. As a result of these combined changes, this column of cells will deliver less total Notch signal in *Lfng/Mfng* double mutants (see *Figure 4A*), and thus the adjacent cell column in Kölliker's organ also differentiates into inner hair cells and inner phalangeal cells. We summarize the effects and mechanism of mutating different components of the Notch pathway in *Figure 7*.

Our results suggest that the first phase of Notch signaling at the boundary of the organ of Corti establishes a single column of cells destined to form both inner hair cells and inner phalangeal cells. As this column of *Atoh1, Lfng, Mfng, Jag2* and *Dll1*-expressing progenitors develops, we propose that a second fate decision between an inner hair cell and an inner phalangeal cell is achieved by conventional hair cell-supporting cell lateral inhibition mediated through Dll1 and Jag2 (*Figure 6C*, green cells; *Hicks et al. (2000)*; *Van de Walle et al., 2011*). Our current data, previous expression studies (*Chen et al., 2002*; *Cai et al., 2013*) and previous *Atoh1* lineage tracing showing inner phalangeal cells derived from *Atoh1*-expressing cells (*Yang et al., 2010*) suggests this entire column of cells initially expresses early hair cell genes such as *Atoh1* and *Mfng.* Since we only observe a conversion of inner phalangeal cells to inner hair cells when Notch signaling is strongly reduced (in Notch1 mutants or at high doses of Notch inhibitors; *Figures 3* and *5*), we suggest this second phase of Notch signaling involves high levels of Notch signaling and is resistant to subtle manipulation of Notch signals. This sequence of organ of Corti differentiation is concluded as inner hair cells begin to express Fgf8, which causes cells lateral to the inner hair cells (*Figure 6C*, green hair cell) and inner phalangeal cells (*Figure 6C*, yellow cell) to differentiate as pillar cells (*Mueller et al., 2002*; *Jacques et al., 2007*).

This model for formation of the neural border of the organ of Corti explains the observations reported here, together with data from previous studies. The moderate level of Notch signaling delivered at the border of the prosensory domain and Kölliker's organ has been visualized in previous studies (*Murata et al., 2006*; *Basch et al., 2011*; *Liu et al., 2013*; *Figure 4A*). Indeed, Murata and colleagues have observed Atoh1-expressing cells adjacent to cells containing activated Notch1 at the boundary of Kölliker's organ (*Murata et al., 2006*). We show that reducing this moderate Notch signaling leads to the formation of additional columns of cells fated to form inner hair cells and inner phalangeal cells. Although this phenotype can be observed in a number of previous studies in which Notch signaling was reduced or particular Notch ligands or Notch targets were mutated (*Lanford et al., 1999*; *Zheng et al., 2000*; *Zine et al., 2001*; *Kiernan et al., 2005*; *Brooker et al., 2006*; *Tateya et al., 2011*; *Abdolazimi et al., 2016*), the duplication of inner phalangeal cells in these studies was not identified due to the lack of specific markers for these cells. It may be necessary to re-evaluate these phenotypes in light of the present study.

Our model can also explain why mutation of *Lfng* alone is able to rescue the inner hair cell phenotype, but not the outer hair cell phenotype of *Jag2* mutant mice (*Zhang et al., 2000*). Since Lfng normally enhances *cis*-inhibition between Delta ligands and Notch receptors expressed in the same cell (*LeBon et al., 2014*), we predict that loss of *Lfng* would partially relieve *cis*-inhibition of Dll1 expressed at the boundary and allow it to deliver more Notch signal to compensate for the loss of *Jag2*. We predict that *Mfng* mutants would rescue *Jag2* mutant mice in a similar manner. Although *Fringe* genes are often expressed in overlapping patterns, no obvious synergy has been reported in the brain, axial skeleton, limbs or cranial nerves for mutants in multiple family members (*Moran et al., 2009*). However, an interaction between *Lfng, Rfng* and *Mfng* has been observed in the maturation of marginal zone B cell precursors (*Tan et al., 2009*; *Song et al., 2016*), suggesting that in at least some cases, the combined effects of these three enzymes on Notch signaling in either *cis* or *trans* cannot be achieved by either alone.

We found no strong evidence for the supernumerary inner hair cells and inner phalangeal cells being caused by an increase in proliferation. Our EdU labeling comparisons between wild type, *Pofut1* and dnMAML1 mutant mice showed no significant increase in labeling of inner phalangeal cells, inner hair cells or any other organ of Corti cell type. A previous study described significant increases in proliferation in the progenitors of pillar cells, Deiters' cells and Hensen's cells in $Dll1^{+/-}$; $Jag2^{-/-}$ and $Dll1^{hyp/-};Jag2^{-/-}$ mutant embryos, but not single mutants for either ligand (*Kiernan et al., 2005*). We infer from this that the reduction in Notch signaling in all our Notch

pathway mutants (with the exception of *Notch1* conditional mutants) is insufficient to trigger aberrant proliferation, and that our phenotype is due solely to a disruption in the boundary between the prosensory domain and Kölliker's organ.

It is now well established that the strength of Notch signaling in vertebrates can vary according to both the identity and post-translational modification of the activated Notch receptor and the identity of the activating ligand (for example, *Ong et al., 2006*; *Yamamoto et al., 2012*; *Lee et al., 2013*; *Van de Walle et al., 2013*; *LeBon et al., 2014*; *Petrovic et al., 2014*; *Gama-Norton et al., 2015*). The developing organ of Corti is an excellent system to study the strength of Notch signaling, as its stereotyped pattern of hair cells and different supporting cell types allows extremely small irregularities in cell numbers to be detected easily. Our data provide an example of how rapid dynamic changes in both Notch ligands (Jag1, Jag2, and Dll1) and regulatory Fringe proteins at the boundary of the organ of Corti can quickly transition between Notch signaling states that set the boundary (moderate Notch signaling; regulated by Fringe proteins) and then distinguish between hair cells and supporting cells (high Notch signaling; no requirement for Fringe proteins). Moreover, our observation that the early phase of Notch signaling is extremely sensitive to changes in signaling strength suggests that the Hes or Hey genes activated in this phase will have mostly low-affinity binding sites for the Rbpj transcriptional activator in their promoters. Indeed, low levels of Notch activation and Hes/Hey gene expression are observed at this boundary (*Murata et al., 2006*; *Basch et al., 2011*; *Tateya et al., 2011*) and *Hes5*, whose promoter contains low-affinity Rbpj binding sites (*Ong et al., 2006*), is up-regulated at the boundary of Kölliker's organ as hair cells differentiate (*Tateya et al., 2011*). Furthermore, *Hes5* mutant mice develop supernumerary inner hair cells and supporting cells at the onset of hair cell differentiation (*Zine et al., 2001*).

Our results provide additional evidence for the notion that organ of Corti development is an iterative process proceeding in the neural-abneural direction, with inner hair cells and their associated supporting cells differentiating first, which then release inducing signals such as Fgf8 that then drive the differentiation of the adjacent cells into pillar cells. It is likely that these short-range signals interact with and modify other patterning signals in the cochlear duct, such as FGF10 and FGF20 (*Huh et al., 2012*; *Urness et al., 2015*), although signals that promote specific cell fates in the outer hair cell region, such as outer hair cells and Deiters' cells, remain to be identified. We do not know the nature of signals that limit the size of the organ of Corti on the abneural side of the cochlear duct, but given the proximity of a strong source of BMP4 in the future outer sulcus (*Morsli et al., 1998*; *Ohyama et al., 2010*), it is likely that this factor may play a role. Nevertheless, our data reveal how the deployment of the Notch pathway-modifying enzymes Lfng and Mfng at the boundary of the future organ of Corti modulate levels of Notch signaling to restrict hair cell-inducing signals to a precise location at this boundary. Since Lfng and Mfng expression subsequently diverge, with Mfng restricted to hair cells (*Cai et al., 2015*) and Lfng restricted to supporting cells (*Zhang et al., 2000*), it will be of interest to determine whether these enzymes continue to play any role in modulating Notch signaling in the mature organ of Corti.

## Materials and methods

### Experimental animals

Double homozygous *Lfng* and *Mfng* mutant embryos were generated by crossing *Mfng* homozygous mutant mice (B6(FVB)-*Mfng*^tm1.1Cfg^/Mmucd; RRID:MGI:5615604) obtained from the Functional Glycomics Consortium; http://www.functionalglycomics.org and available at the MMRRC, stock number 031948-UCD) with heterozygous *Lfng* mice (B6;129S1-*Lfng*^tm1Grid^/J; RRID:IMSR_JAX:010619) from the Jackson Laboratory (stock number 010619), and crossing the resulting compound mutant offspring. Triple homozygous *Lfng;Mfng;Rfng* embryos were obtained from Dr. Susan Cole, Ohio State University. Inner ear-specific dominant negative Mastermind-like (*dnMAML1*) mice, were generated by crossing *Pax2-Cre* mice (*Ohyama and Groves, 2004*) with a mouse carrying a human dnMAML construct downstream of a floxed PGK-neo-tpA cassette targeted to the ROSA26 locus (*Tu et al., 2005*). Mice homozygous for conditional alleles of *Pofut1* (*Shi et al., 2005*) were crossed with *Pax2-Cre* mice that were also heterozygous for a null mutation in *Pofut1*. *Pax2-Cre* mice are available from the MMRRC (stock number: 010569-UNC; RRID:MGI:4438962). *Glast-DsRed* reporter mice (*Regan et al., 2007*) were a kind gift from Ben Deneen at Baylor College of Medicine with the

permission of Jeffrey Rothstein. They were crossed to Atoh1$^{A1GFP/A1GFP}$ reporter mice (*Shroyer et al., 2007*); RRID:IMSR_JAX:013593) to generate the embryos used in organotypic cultures. At least four cochleas were analyzed for each genotype at each age. The following primers were used for genotyping:

*Pax2-Cre* allele: Cre1F (GCCTGCATTACCGGTCGATGCAACGA), Cre1R (GTGGCAGA TGGCGCGGCAACACCATT) yield a 700 bp band.

*Pofut1* floxed, deleted and wild type allele: Forward primer GGG TCA CCT TCA TGT ACA AGT GAG TG and reverse primer ACC CAC AGG CTG TGC AGT CTT TG yield a 960 bp floxed allele band, and either a 700 bp wild type band or a 300 bp deleted allele band.

*Manic fringe* deleted and wild type allele: Forward primer GTG CTG AAG CAG AGG CCA TG and reverse primer CAA GGT GAA GGA GCC CAG TT yield a 370 bp band for the deleted allele; forward primer GGC CCT CTC TCA CAT GGA TTT T and reverse primer TCT ACC TCC AAG CAC TAA GG yield a 444 bp band for the wild type allele

*Lunatic fringe* deleted and wild type allele: Forward primer CCA AGG CTA GCA GCC AAT TAG and reverse primer GTG CTG CAA GGC GAT TAA GTT yield a mutant band of 450 bd; forward primer GAA CAA ATA TGG CCA TTC ACT CCA and reverse primer GGT CGC TTC TCG CCA GGG CGA yield a wild type band of 450 bp.

## Organotypic cochlear cultures

Cochleas from stage E14.5 embryos were collected in PBS and incubated in calcium-magnesium free PBS containing dispase (1 mg/ml) and collagenase (1 mg/ml) for 8 min at room temperature as previously described (*Doetzlhofer et al., 2009*) to free the cochlear duct from surrounding condensed mesenchyme tissue. Embryonic cochlear explants were cultured on SPI black membranes (SPI Supplies) in DMEM-F12 (Invitrogen) with N2 supplement and Fungizone. All cultures were maintained in a 5% $CO_2$/20% $O_2$ humidified incubator for 48 or 72 hr. Cultures were treated with function blocking antibodies against IgD or against the Notch1receptor (*Wu et al., 2010*) provided by Genentech or with DMSO and the gamma secretase inhibitor IX (DAPT). For cell counts, three cochleas were counted per condition for each replicate, with at least four independent biological replicates performed. Cultures were excluded from analysis if they showed signs of contamination or if the cochlear explant failed to attach or displayed grossly abnormal morphology.

## In situ hybridization

E14.5, E16.5 or P0 inner ears were fixed in 4% paraformaldehyde in PBS overnight at 4°C, sunk in 30% sucrose in PBS at 4°C, incubated in OCT at room temperature for 1 hr, and frozen in liquid nitrogen. Digoxigenin-labeled antisense riboprobes to mouse *Lunatic fringe*, *Manic Fringe* and *Atoh1* were synthesized using standard protocols (*Stern, 1998*). In situ hybridization was performed as recently described (*Cai et al., 2015*). Fluorescent in situ hybridization was performed according to a modified protocol (*Denkers et al., 2004*). Slides were incubated in DEPC-PBS with 3 µl $H_2O_2$/10 ml for 15–30 min, washed three times in DEPC-PBS for 5 min each, dried and then incubated in hybridization buffer containing the FITC- and/or DIG-labeled probes overnight at between 65–70°C. The following day, slides were washed for 10 min in 0.2x SSC at 65–70°C, and then twice more for 25 min each in 0.2x SSC at 65–70°C. Slides were then washed at room temperature for 5 min in MABT and then blocked for 30–6- minutes in MABT +20% sheep serum +2% Roche Blocking Reagent. Slides were then overnight at 4°C with 1:500 anti-DIG-HRP or anti-FITC-HRP antibodies in blocking buffer. The following day, the slides were washed three times in MABT for 5 min each, then incubated for 30 min in Tyramide working solution (TSA plus kit, Perkin-Elmer: 10 µl of stock in 500 µl of diluent) and the reaction monitored after 30 min by fluorescence microscopy until the desired staining was observed. If a second color is desired, slides were then washed and the antibody staining and Tyramide amplification repeated with anti-DIG-HRP or anti-FITC-HRP antibodies..

## Immunohistochemistry

Antibodies used in this study were anti-p27$^{Kip1}$ (NeoMarker/ThermoFisher; RRID:AB_1959178), anti-myosin-VI (Proteus; RRID:AB_10013626), anti FABP7 (a kind gift from Drs Yuji Owada and Nobuko Tokuda, Yamaguchi University, Japan), anti GLAST (Militenyl Biotec; RRID:AB_10829302), Nerve Growth Factor Receptor (p75, Advanced Targeting Systems; RRID:AB_171798), anti Jagged1 (Santa

Cruz; RRID:AB_649685) and anti cleaved Notch1 (Cell Signaling; RRID:AB_331612). Alexa 594-conjugated phalloidin (Thermo Fisher; RRID:AB_2315633) was used to label actin in hair cell stereocilia. Secondary antibodies used were Alexa-594 or Alexa-488 (Thermo Fisher) and anti-rabbit HRP (Thermo Fisher). DAB staining was done using a DAB kit (Vector labs). For anti-p27$^{Kip1}$ and Jagged1 staining, sections were boiled for 10 min in 10 mM citric acid pH 6.0. For anti-GLAST and anti-N1ICD staining, paraffin sections were boiled for 10 min in a pressure cooker with unmasking solution (Vector labs) as described in *Morimoto et al. (2010)*. The signal was enhanced using an ABC kit (Vector Labs) and TSA kit (Perkin Elmer).

## Cell proliferation assay

Timed pregnant females were injected stating at E14.5 with 1 mg of EdU diluted in PBS for every 20 grams of weight. Injections were performed three times a day every 4 hr for three consecutive days. At E18.5, embryos were collected, fixed in 4% PFA and cryosectioned. EdU detection was performed according to the manufacturer's instructions with a ClickiT EdU Alexa Fluor 594 system (Thermo Fisher). Sections were also stained with Myosin VI antibodies and counterstained with DAPI. Positive cells from controls and mutant embryos were counted blind and significance was established using a modified Wald test (*May and Johnson, 1997*).

## Imaging and measurements

Images were taken using a Zeiss Axiophot microscope, or a Zeiss Axio Observer fluorescent microscope with an Apotome structured illumination attachment. Images were processed using Axiovision software then further processed in Adobe Photoshop. Inner and outer hair cells were counted in dissected or cultured cochleas per 200 mm using Axiovision and Adobe Photoshop software. Cochlea and cochlea culture lengths were measured using Axiovision software.

## RNA extraction and Real-Time PCR

For RNA extraction, three cochlear cultures were pooled and total RNA was isolated by using a QIAGEN RNeasy Micro kit. cDNA was synthesized by using SuperScript III Reverse Transcription Reagents (Invitrogen). qPCR was performed with a Power SYBR Green kit (Applied Biosystems) and gene-specific primer sets on a StepOnePlus real-time PCR system (Applied Biosystems). Three technical replicates were performed for each qPCR reaction. At least four biological replicates of three cochleas each were performed for each genotype or culture condition. Relative gene expression was analyzed by using the $\Delta\Delta$CT method (*Livak and Schmittgen, 2001*) and compared with Student two-tailed t test. cDNA from untreated cochlear explants was used as a calibrator, and a ribosomal gene (L19) was used as endogenous references. Gene-specific primer sets used are: Atoh1: F 5'-ATGCACGGGCTGAACCA-3'; R 5'-TCGTTGTTGAAGGACGGGATA-3', L19: F 5'-GGTCTGGTTGGATCCCAATG-3'; R 5'-CCCGGGAATGGACAGTCA-3', Hes5: F 5'-GCACCAGCCCAACTCCAA-3'; R 5'-GGCGAAGGCTTTGCTGTGT-3', Hey1: F 5'-CACTGCAGGAGGGAAAGGTTAT-3'; R 5'-CCCCAAACTCCGATAGTCCAT-3', Hey2: F 5' AAGCGCCCTTGTGAGGAAA-3'; R 5'-TCGCTCCCCACGTCGAT-3', Heyl: F 5'-GCGCAGAGGGATCATAGAGAA-3'; R 5' TCGCAATTCAGAAAGGCTACTG-3', Hes1: F 5'-GCTTCAGCGAGTGCATGAAC-3'; R 5'-CGGTGTTAACGCCCTCACA-3'.

## Acknowledgements

We thank Hongyuan Zhang, Alyssa Crowder, Huiling Li and Bridgett McNulty for excellent technical assistance. We thank Jeffrey Rothstein for *GLAST-dsRed* transgenic mice, Huda Zoghbi for *Atoh1$^{A1GFP}$* knock-in mice, Warren Pear for *dnMAML1* mutant mice and Pamela Stanley and Shaolin Shi for *Pofut1* conditional mutant mice. We thank Doris Wu for the *Lfng* probe and Yuji Owada and Nobuko Tokuda for their gift of FABP7 antibodies. This work was supported by DC006185 (AKG)

# Additional information

## Funding

| Funder | Grant reference number | Author |
|---|---|---|
| National Institute on Deafness and Other Communication Disorders | NIH DC006185 | Andrew K Groves |

The funders had no role in study design, data collection and interpretation, or the decision to submit the work for publication.

## Author contributions

MLB, Conception and design, Acquisition of data, Analysis and interpretation of data, Drafting or revising the article; RMB, H-IJ, Acquisition of data, Analysis and interpretation of data, Drafting or revising the article; FS, CRN, Acquisition of data, Contributed unpublished essential data or reagents; FD, Conception and design, Contributed unpublished essential data or reagents; RKE, HZ, Acquisition of data, Analysis and interpretation of data; TG, SEC, AD, MM-S, Drafting or revising the article, Contributed unpublished essential data or reagents; NS, Conception and design, Drafting or revising the article, Contributed unpublished essential data or reagents; AKG, Conception and design, Analysis and interpretation of data, Drafting or revising the article

## Author ORCIDs

Andrew K Groves, http://orcid.org/0000-0002-0784-7998

## Ethics

Animal experimentation: This study was performed in strict accordance with the recommendations in the Guide for the Care and Use of Laboratory Animals of the National Institutes of Health. All animal experiments in this study were carried out in accordance with the Institutional Animal Care and Use Committee protocol (AN4956) at Baylor College of Medicine.

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
