## [Decision Letter]

Thank you for submitting your article "Fringe proteins fine-tune Notch signaling to set the boundary of the organ of Corti and establish sensory cell fates" for consideration by *eLife*. Your article has been favorably evaluated by Marianne Bronner (Senior Editor) and three reviewers, one of whom, Tanya Whitfield, is a member of our Board of Reviewing Editors. The following individual involved in review of your submission has agreed to reveal his identity: Fernando Giraldez (Reviewer #2).

The reviewers have discussed the reviews with one another and the Reviewing Editor has drafted this decision to help you prepare a revised submission.

This is an interesting study that addresses the role of Fringe family proteins in regulating Notch signalling in the developing organ of Corti of the mammalian inner ear. Notch signalling is known to pattern the organ of Corti, and Fringe proteins are also known for their role in fine-tuning Notch signalling in other systems. The new finding here is that regulation of Notch signalling by Fringe plays a role in the generation of inner hair cells and their associated supporting cells, the inner phalangeal cells, in the cochlear duct. A surprising result is that a partial loss of Notch signalling results in both supernumerary hair cells and supernumerary supporting cells. This differs from the classical model for lateral inhibition in the developing cochlea, where a disruption of Notch signalling is predicted to result in supernumerary hair cells at the expense of supporting cells. The data are clear and well-presented, with careful and appropriate quantitation, use of controls and statistical analysis.

All three reviewers were positive towards the work and found it interesting, novel and potentially important. They praise the careful approach and high quality of the data presented. However, there was consensus that the manuscript needed revision in places, especially with respect to the focus on cis-inhibition. The revisions must therefore include:

1) Improved clarity of the explanations of cis-inhibition, and the relative roles of Fng proteins and DSL ligands in regulating Notch activity.

2) The argument that levels of Notch activity differ in different cell types and regions of the cochlear duct is critical to the model. Further experimental support for the claim that Notch signalling activity levels differ both over time and in different cell types in the developing cochlear duct is required (see comments from reviewers 2 and 3).

3) Further experimental support for cis-inhibition and the effects of Fng on cis-inhibition in the developing cochlear duct is required. If this cannot be provided, the argument for cis-inhibition should be presented as an untested model and the overall focus on cis-inhibition in the manuscript should be toned down.

4) Strengthen the claim that Lfng marks developing inner hair cells, as suggested by reviewer 3.

5) Addressing additional comments in the following reviews is at the authors' discretion.

Reviewer #1:

1) The authors show that they can phenocopy the Fng mutant phenotype with the Jag2 mutant. The information presented in the Introduction indicates that Fng acts differentially on Dl-N and Jag/Ser-N interactions, whereas in the first paragraph of the subsection “Reduction in Notch signaling leads to the formation of supernumerary inner hair cells and inner phalangeal cells”, it is stated that Fng can potentiate signalling through Notch from both Dll1 and Jag2. It would therefore be interesting to know whether a Dll1 phenotype also phenocopies the Fng and Jag2 results. I appreciate that this may be beyond the scope of the current study, but would the authors predict a similar or a different result?

2) If Fng proteins are required for precision fine-tuning of Notch signalling, why is the system so robust to perturbations in one or the other Fng, and a phenotype is only seen when the functions of both genes are lost?

3) I am not really sure why the authors propose that their findings argue for a new role for Notch (Discussion, first paragraph); an interpretation could be that this is just another illustration of lateral induction, albeit in a restricted zone of the cochlea. Can the authors provide more information as to why they do not think this is the case?

4) Is extension of the cochlea normal? Are there more cells overall, or are they packed in a different way due to altered shape of the organ?

5) In trying to get to grips with the terminology here, I find that the terms 'lateral inhibition' (involving activation of Notch signalling) and 'cis-inhibition' (involving attenuation of Notch signalling) are potentially confusing/ambiguous and difficult to follow.

For example, the following statement is made in the Introduction: 'In vertebrates, Lfng and Mfng proteins regulate receptor-ligand interactions in cis in the same way as they do in trans: they increase cis-inhibition between Notch and Δ ligands, while attenuating cis-inhibition between Notch and Serrate/Jagged ligands (LeBon et al. 2014)'.

The wording 'in the same way' here is confusing, because it appears to be the opposite to the information given in the second paragraph of the Introduction, where it is stated that Fng proteins increase the level of Notch signalling by Δ. If Fng increases cis-inhibition between N and Dl (later in the aforementioned paragraph), given the information presented, that should lead to a decrease in Notch signalling activity in the Notch-expressing cell.

Likewise, the statement that Fng proteins 'attenuate Notch signaling by Serrate/Jagged ligands' (Introduction, second paragraph) appears to be contradictory to that later in the aforementioned paragraph where Fng proteins function by 'attenuating cis-inhibition between Notch and Serrate/Jagged ligands' (which in other words would potentiate Notch signalling via Ser/Jag).

I appreciate that the terms 'lateral inhibition' and 'cis-inhibition' are now entrenched in the literature, but I, for one, find the explanations above contradictory and feel that more clarification is needed. If the authors can make sure that their text is completely unambiguous that would be helpful.

6) I don't see how the descriptions of Fng activity in the Introduction (second paragraph) are illustrated in the summary diagram (Figure 6). If Fng potentiates Dl-N cis-inhibitory interaction, but attenuates Jag-N cis-inhibitory interaction, the dotted arrow from Fng to Jag-N in the green cell should be drawn as an inhibitory line (-|), not an arrow, and the cis-inhibitory interactions between Dl and N should be stronger (thicker lines) in the green cell than those in the white cell, whereas the diagram shows the opposite. These apparent discrepancies make it hard to follow the argument.

7) In the legend to Figure 6, the statement 'As a result, these cells do not respond to hair cell-inducing signals' seems to contradict the sentence at the beginning of the section describing panel B, which states that these cells are responding to hair cell-inducing signals, and have been shaded green to indicate this. If the sentence refers to the neighbouring cells, the 'these' is ambiguous, and the sentence does not follow logically from the previous one. If the authors mean that the green cells have responded to hair cell-inducing signals, but have not yet received instruction to undergo final differentiation into hair cells (as in C), this needs clarifying further.

Reviewer #2:

The paper by Basch et al. is a very interesting analysis of the function of Fringe proteins in hair cell patterning in the ear. The main and certainly interesting aspect of the work is the disclosure of a very specific function of Fng proteins in cochlear development and the exploration of their association with Notch signalling. The experiments are extremely careful and beautifully presented, and there is no doubt about the consistency of the various phenotypes and treatments. Data are of high quality and involve a variety of approaches including lineage tracing, phenotypic analysis, organ culture, etc., all making it a robust piece of work. The authors propose an interesting model that involves Notch activity levels and cis- and trans- modes of ligand operation to accommodate the results and with those of the literature. However, it is here where the work requires some improvement.

1) Figure 1 shows that the expression of Lfng and Mfng are restricted to sensory patch. Although the expression patterns are overlapping, they are quite dissimilar. This suggests that modulation of Notch is different in the two cell types. Could it be that Lfng and Mfng have different effects on different ligands? Is it possible that Lfng maintains a low and constant Notch activity and Mfng favours Dll signalling?

No doubt about the specificity of the combination of Lfng+Mfng for the correct development of the inner row. However, the correlation with Notch signalling levels remains obscure. Lfng+Mfng phenotype gives an expanded inner cell row, and this is paralleled by the loss of Jag2, but also of Jag1. Jag1 is expressed all throughout prosensory development and in supporting cells, but Jag2 is expressed along with Dll1 only in hair cells. How this is matched?

2) Lineage tracing. Those are interesting experiments and show that almost the complete sensory epithelium derives from Lfng expressing progenitors (probably also from Jag1-positive ones). However, genetic tracing shows the common origin of the sensory epithelium, but not much about the restricted expression. In this view, the genetic tracing of Manic cells would have been of interest, because it seems that the uniqueness of the effects may come from the co-expression.

3) Fng LOF is mimicked by low Notch. That is the main line of the argument along the paper. But from here, I would expect the authors to consider how this could happen by showing the effects of Fng on ligands or on Notch activity would have been informative. Neither it is clear to me whether the levels of Notch affect the expression of modulators or vice versa. We do not really know how Notch operates outside lateral inhibition and all possibilities should remain open.

4) As to the levels of Notch activation, indeed, the observations fit well with the Murata paper. But nevertheless, the different conditions generated by the experiments in the paper open some questions about what are the levels of Notch in the different regions of the sensory epithelium. In this sense, conditions are not strictly comparable. Mfng and Lfng show overlapping but different expression domains, and in MAM and Pofut CKO or DAPT experiments, the decrease of Notch is ubiquitous. Can this be assessed specifically for Fng mutants? Please comment on that.

5) The data in the bar diagrams of Figure 5 indicate that there is an inverse correlation between Notch activity, Hes/Hey expression and the production of hair cells. And it shows the differential sensitivity of inner and outer hair cells to Notch. This is in good correspondence with MAMl and Pofut experiments. But what about in Fng mutants? It would have been interesting to see Hes/Hey levels in Fng mutants.

6) Cis-inhibition: The experiments say little about cis-inhibition. "Lfng activity attenuates Jag1-Notch signaling between cells (Hicks et al. 2000; Rana and Haltiwanger 2011) and promotes Jag1-Notch cis-inhibition in the same cell, with the result…" I am not sure how the second sentence follows the first and, moreover, whether it is a general true at all. Is there a good evidence for cis-inhibition in ear sensory cells? The results from Daudet's lab indicate rather the opposite for Dll1.

The argument on cis-inhibition continues in the second and third paragraphs of the Discussion, quoting LeBon 2014 as if an ear paper…: "we hypothesize that this column of cells only delivers a moderate level of Notch signaling, due to active cis-inhibition between the Notch ligands and Notch receptors that can occur in the same cell". However, the experiments do not provide any evidence of that occurring in the ear…

In summary, the results are indeed sound and of general interest, but as to the model, it is unclear that the function of Fng on Notch and cis-inhibition are sufficiently substantiated. Leaving aside cis-inhibition, which actually I see out of the paper, the model still needs a link between Fng, ligands and Notch in the cochlea. Something that may give a hint of what are the effects of Fng on Notch, Notch ligands or Fng. The main evidence for the model is correlative, the parallel between phenotypes. The main question, in my view, is what is the connection between both.

Reviewer #3:

The organ of Corti is characterized by strikingly precise rows of hair cells, although how this precision is established is not known. This manuscript by Basch et al., proposes a novel form of Notch signaling that sets up the boundary of the initial row of inner hair cells, in which Fringe proteins, modulators of Notch signaling, are an essential component. The authors show that Lfng and Mng are initially precisely positioned at the location of the first inner hair cell row. In their absence, multiple rows form, along with a duplication of supporting cells. By comparing the results of minor reductions in Notch signaling to more robust loss of Notch, the authors suggest that establishment of the initial row of hair cells is inherently different in that it requires lower levels of Notch signaling, and is therefore more susceptible to mild reductions in Notch signaling. Overall the data is well-performed and presented, and the authors analyze quite a number of different Notch mutants to determine differences. The authors put forward an interesting hypothesis as to why the hair cell genesis is initially restricted to a single row. However, it is not clearly established that a lower level of Notch signaling is required for boundary formation – while milder defects in Notch signaling may first lead to an expansion of the sensory domain, it may be that the non-sensory regions (or supporting cells outside the hair cell regions) are simply more sensitive to loss of Notch signaling, and thus boundary formation is lost first.

The authors suggest that cis-inhibition mediated by Lfng and Mfng expression results in moderate levels of Notch activation in nearby Kolliker's cells (Figure 6), and that this is important for setting up the boundary. However, in this case the prediction would be that deleting both fringes would lead to higher levels of Notch, as cis-inhibition would be largely relieved. However, this does not seem to be the case, as new hair cells develop, suggesting Notch activity is reduced in their absence. Since the model indicates that moderate levels of Notch activity are important for setting up this boundary, it would be important to show directly how levels of Notch are regulated at the boundary.

It is not clear that severe loss of Notch signaling shows an inherently different phenotype than milder forms. Is it not more likely that the severe loss obscures the initial milder phenotype? After all, there is a dramatic loss of boundary formation in the severe cases of loss of Notch signaling. While additional supporting cells are present in the milder cases, these may be induced by the excess hair cells, which themselves also convert to hair cells in the case of more severe reductions in Notch.

If the regulation of inner hair cell formation to a single row is not via lateral inhibition, what type of signaling is it? It would seem that even at the proposed lower levels the cell expressing the ligand is inhibiting nearby cells (in Kolliker's organ) from adopting the hair cell fate, and therefore this would fit the definition of lateral inhibition (see Figure 6).

The authors show that Lfng likely initially marks the inner hair cell using fate-mapping, although given that they have reporters for both Lfng and Atoh1-it seems relatively straightforward to show that these both overlap during differentiation, further strengthening the hypothesis that Lfng and Mfng mark the inner hair cell boundary. It would also be interesting to look at the onset of these factors-Does Lfng mark the inner hair cell prior to Atoh1? Or vice versa?

The suggestion that Lfng modulates Jag1 levels initially (Figure 6) has not been demonstrated previously-is there any evidence of this from the Lfng loss of function? If not this part of the model should be modified.

---

## [Author Response]

[…] All three reviewers were positive towards the work and found it interesting, novel and potentially important. They praise the careful approach and high quality of the data presented. However, there was consensus that the manuscript needed revision in places, especially with respect to the focus on cis-inhibition. The revisions must therefore include:

1) Improved clarity of the explanations of cis-inhibition, and the relative roles of Fng proteins and DSL ligands in regulating Notch activity.

It may be helpful to explain how we came to think about cis-inhibition of Notch signaling in our study. We had spent a number of years characterizing the cochlear phenotypes of a number of different Notch mutants. In mutant after mutant, we kept observing a duplication of inner hair cells AND inner phalangeal cells, which did not fit in with the standard lateral inhibition model of Notch signaling that distinguishes between hair cells and supporting cells. The common thread in all our mutants was that they reduced, but did not completely abolish, Notch signaling. We were also able to confirm this phenotype in culture by quantitatively reducing Notch signaling with small doses of different Notch inhibitors.

Our results are consistent with a model in which a column of differentiating progenitor cells that express Atoh1, Jag1, Jag2 and Dll1 is induced at the boundary of the future organ of Corti. This column of cells sends a Notch signal to the adjacent cells in Kölliker’s organ and stops them adopting the same Atoh1+ fate. Thus, deleting Notch ligands, or genes that are required for receiving Notch signals reduces this inhibitory signal and causes a second column of cells to form, duplicating both the inner hair cells and the inner phalangeal cells.

In the course of this work, we realized that this column of boundary cells was transiently marked by expression of Lfng and Mfng. We verified this by fate mapping with our Lfng-CreER mice. When we examined Lfng/Mfng double knockouts, we observed the same duplication phenotype seen in all our other Notch pathway mutants. We have now also examined activated Notch protein levels and find reduced Notch signaling in these double mutants. Our Lfng and Mfng expression data and double knockout results are therefore consistent with Lfng and Mfng acting in this column of cells to promote Notch signaling from this column to adjacent cells. The question is: how do they do this?

In principle, there are two mechanisms by which Fringe proteins can regulate the amount of Notch signal delivered by a cell. One is direct glycosylation of Notch ligands by Fringe, rather than Notch receptors. Although this has been shown to occur in vitro, there is very little published data on the biological relevance of these modifications, and what data there is suggests that Fringe modification of the ligands is not necessary for their function. The second mechanism is the regulation of cis-inhibition by Fringe proteins between Notch receptors and ligands in the same cell, which reduces the activity of both receptor and ligand. The best evidence for cis-inhibition and the role of Fringe proteins in this mechanism comes from *Drosophila* genetics and cell line studies rather than from in vivo vertebrate studies, but the phenomenon is nevertheless well accepted in the literature.

There is no reliable marker to detect cis-inhibition, as it is simply an attenuation of Notch signaling under certain circumstances. We therefore stress that we are invoking cis-inhibition as an *explanation* for our experimental data, rather than observing it directly. We are happy to consider other explanations if the reviewers can suggest some, but cis-inhibition is the best explanation we have for the Lfng/Mfng double mutant phenotype at present. We have tried to explain the phenomenon of cis-inhibition more clearly in the revised manuscript. We have also included more data on the amount of Notch signaling seen in Lfng/Mfng mutants to support our explanation. Finally, in recognition of the fact that much of our data on loss of function mutants does not involve Fringe activity directly, we have revised the title of the paper to focus more on Notch signaling and less on Fringe function.

2) The argument that levels of Notch activity differ in different cell types and regions of the cochlear duct is critical to the model. Further experimental support for the claim that Notch signalling activity levels differ both over time and in different cell types in the developing cochlear duct is required (see comments from reviewers 2 and 3).

We have now added data showing how levels of Notch signaling change in Lfng;Mfng double knockouts and how this is reflected by changes in cell fate only at the border of the organ of Corti. We hope that this data, together with the demonstration from multiple mouse knockouts and culture experiments that different regions of the cochlea are differentially affected by reduction of Notch signaling demonstrates that Notch activity differs in different regions of the cochlea.

3) Further experimental support for cis-inhibition and the effects of Fng on cis-inhibition in the developing cochlear duct is required. If this cannot be provided, the argument for cis-inhibition should be presented as an untested model and the overall focus on cis-inhibition in the manuscript should be toned down.

As mentioned above, we have now reduced our emphasis of this phenomenon in the revision and have clarified our explanation of its effect on Notch signaling as it relates to the Lfng/Mfng phenotype. We would emphasize that there is no *specific method* of detecting cis-inhibition, other than observing changes in the strength of Notch signaling after manipulations that are predicted to increase or reduce cis-inhibition. We have now provided more data on Notch signaling strength in Lfng/Mfng mutants in the revised manuscript.

4) Strengthen the claim that Lfng marks developing inner hair cells, as suggested by reviewer 3.

We have shown that Lfng, Atoh1 and Mfng co-localize transiently in the cochlea and that some Lfng-expressing cells go on to form inner hair cells by using Lfng-CreER mice. These events happen before more mature hair cell markers (such as Myosin7a) are expressed, and so we feel this is the best data we can propose in support of this argument. We discuss this issue further in our response to reviewer 3, below.

5) Addressing additional comments in the following reviews is at the authors' discretion.

We have added responses to all the comments from the reviewers below.

Reviewer #1:

1) The authors show that they can phenocopy the Fng mutant phenotype with the Jag2 mutant. The information presented in the Introduction indicates that Fng acts differentially on Dl-N and Jag/Ser-N interactions, whereas in the first paragraph of the subsection “Reduction in Notch signaling leads to the formation of supernumerary inner hair cells and inner phalangeal cells”, it is stated that Fng can potentiate signalling through Notch from both Dll1 and Jag2. It would therefore be interesting to know whether a Dll1 phenotype also phenocopies the Fng and Jag2 results. I appreciate that this may be beyond the scope of the current study, but would the authors predict a similar or a different result?

We have revised the manuscript to try and resolve this confusion. The *Drosophila* Fringe literature suggests that in most cases characterized to date, Fringe modifications potentiate Δ signaling and attenuate Serrate signaling. In vertebrates, the situation is a more complicated, as there are two Serrate homologues, Jag1 and Jag2 and multiple Δ ligands. Current evidence strongly suggests that Jag1 acts like Serrate in flies: Fringe modification tends to attenuate Jag1 signaling. This was shown most recently and in greatest detail by a recent cell line study in *eLife*:

LeBon et al., “Fringe proteins modulate Notch-ligand cis and trans interactions to specify signaling states”. *eLife*. 2014 Sep 25;3:e02950.

However, there are almost no published data on the effects of Fringe proteins on Jag2 signaling. As the reviewer mentions, we found one study that suggests Lunatic Fringe *potentiates* Jag2 signaling in a similar manner to Δ ligands rather than attenuating it as is the case for Jag1:

Van de Walle et al., “Jagged2 acts as a Δ-like Notch ligand during early hematopoietic cell fate decisions”. Blood. 2011;117(17):4449-59.

Although we cited this for the sake of completeness, the potentiation of Jag2 signaling by Lfng observed by the authors in this study, although statistically significant, was extremely small – an increase in signaling of just 1.15 times. We have therefore decided to remove this citation from the paper. Moreover, unpublished data recently communicated to us by the Elowitz lab (the authors of the 2014 *eLife* paper above) suggests that Lfng and Mfng modification of Notch receptors attenuates both Jag1 and Jag2 signaling. We have therefore revised the manuscript to reflect the best available evidence; namely that Fringe proteins potentiate Δ signaling and attenuate Jagged signaling.

The published work of Kiernan et al. (2005) looking at Dll1 nulls and hypomorphs, and Brooker et al. (2006) looking at Dll1 conditional mutants in the ear suggest that loss of Dll1 will cause a similar phenotype to what we observe in Jag2 mutants. However, as the reviewer notes, we do not have access to the Dll1 conditional allele, and so we cannot perform this study in a timely fashion. Nevertheless, we agree that it is an excellent experiment, and we have included our prediction about the Dll1 conditional phenotype in the revision.

2) If Fng proteins are required for precision fine-tuning of Notch signalling, why is the system so robust to perturbations in one or the other Fng, and a phenotype is only seen when the functions of both genes are lost?

Our data on the expression of Lfng and Mfng shown in Figure 1 suggests that Mfng appears in hair cell progenitors at approximately the same time as Atoh1. Lfng, on the other hand is much more dynamic – it is expressed first in the GER/Kolliker’s organ, then localizes to the same Atoh1-expressing cells as Mfng at the boundary of the organ of Corti, and finally ends up in supporting cells. Thus, there is only one time and place at which the expression of these two Fringe proteins occurs in the same cell type – exactly at the boundary of the organ of Corti as the first Atoh1-expressing hair cell progenitors. It is here that we observe the phenotype reported in our paper. We therefore believe that Lfng and Mfng have redundant functions at this boundary, such that loss of either Fringe gene does not alter the patterning of the boundary.

We do not know why knockouts of either gene alone show no obvious patterning phenotype in other regions of the organ of Corti, but presumably this is because changing Notch signaling by small amounts has no effect elsewhere. That said, it should be noted that we have not examined adult Lfng or Mfng mutant mice, and it is possible that they have subtle phenotypes, such as premature hair cell loss or abnormal hearing thresholds. We have tried to emphasize these points better in the revised manuscript.

We noted in the text that work by Gridley and Kelley (2000) shows that mutation of Lfng can rescue the inner hair cell phenotype of Jag2 mutants, but not the outer hair cell phenotype. This again reinforces the unique sensitivity of the boundary region to Notch signaling. Our results predict that Mfng mutants would rescue Jag2 mutants in a similar manner to Lfng mutants, and we have added this prediction to the text.

3) I am not really sure why the authors propose that their findings argue for a new role for Notch (Discussion, first paragraph); an interpretation could be that this is just another illustration of lateral induction, albeit in a restricted zone of the cochlea. Can the authors provide more information as to why they do not think this is the case?

The reason we believe the column of future inner hair cells and inner phalangeal cells is inhibiting their neighbors though the Notch pathway is that when we *reduce* but not abolish Notch signaling, either by deleting Notch ligands or Notch signaling partners we see an extra row of inner hair cells and inner phalangeal cells. This seems to represent an inhibitory, rather than an inductive interaction.

4) Is extension of the cochlea normal? Are there more cells overall, or are they packed in a different way due to altered shape of the organ?

We have observed no significant differences in cochlear length in any of our mutants. We do see more organ of Corti cells in all our mutants, but this is due to the extra row of inner hair cells and inner phalangeal cells seen when we reduce Notch signaling. We also observe some lateral inhibition defects in the Notch mutants that change the proportion of hair cells and supporting cells, some of which have been previously published (e.g. Jag1 heterozygotes and Jag2 nulls).

5) In trying to get to grips with the terminology here, I find that the terms 'lateral inhibition' (involving activation of Notch signalling) and 'cis-inhibition' (involving attenuation of Notch signalling) are potentially confusing/ambiguous and difficult to follow.

For example, the following statement is made in the Introduction: 'In vertebrates, Lfng and Mfng proteins regulate receptor-ligand interactions in cis in the same way as they do in trans: they increase cis-inhibition between Notch and Δ ligands, while attenuating cis-inhibition between Notch and Serrate/Jagged ligands (LeBon et al. 2014)'.

The wording 'in the same way' here is confusing, because it appears to be the opposite to the information given in the second paragraph of the Introduction, where it is stated that Fng proteins increase the level of Notch signalling by Δ. If Fng increases cis-inhibition between N and Dl (later in the aforementioned paragraph), given the information presented, that should lead to a decrease in Notch signalling activity in the Notch-expressing cell.

Likewise, the statement that Fng proteins 'attenuate Notch signaling by Serrate/Jagged ligands' (Introduction, second paragraph) appears to be contradictory to that later in the aforementioned paragraph where Fng proteins function by 'attenuating cis-inhibition between Notch and Serrate/Jagged ligands' (which in other words would potentiate Notch signalling via Ser/Jag).

I appreciate that the terms 'lateral inhibition' and 'cis-inhibition' are now entrenched in the literature, but I, for one, find the explanations above contradictory and feel that more clarification is needed. If the authors can make sure that their text is completely unambiguous that would be helpful.

To clarify one point for the reviewer (in the opening sentence of point 5), lateral inhibition is the inhibition of *cell fate* that is delivered from one cell to a neighbor. Thus, neurons deploy Notch ligands to inhibit neural progenitors from adopting a neuronal fate, hair cells deploy Notch ligands to inhibit supporting cells from adopting a hair cell fate and so forth. Cis-inhibition, on the other hand, is a *protein-protein interaction* that occurs when a Notch receptor and a ligand are expressed in the same cell and interact with each other. This interaction reduces the amount of free receptor and ligand in the cell, thus lowering the ability of the cell to both send and receive Notch signals. The amount of cis-inhibition that occurs between receptors and ligands in a cell can be regulated by Fringe: Fringe enhances the interaction between Δ and Notch in the same cell (and thus increases cis-inhibition), but decreases the interaction between Jag1/2 and Notch in the same cell (and thus decreases cis-inhibition).

As the reviewer points out, the concept and terminology of cis-inhibition are now embedded in the literature for better or worse. Nevertheless, we are sympathetic to the potential for confusion and have rewritten these parts of the manuscript to try to clarify this issue.

6) I don't see how the descriptions of Fng activity in the Introduction (second paragraph) are illustrated in the summary diagram (Figure 6). If Fng potentiates Dl-N cis-inhibitory interaction, but attenuates Jag-N cis-inhibitory interaction, the dotted arrow from Fng to Jag-N in the green cell should be drawn as an inhibitory line (-|), not an arrow, and the cis-inhibitory interactions between Dl and N should be stronger (thicker lines) in the green cell than those in the white cell, whereas the diagram shows the opposite. These apparent discrepancies make it hard to follow the argument.

As mentioned in point 1 above, we have now tried to clarify what is known about the effects of Fringe on Jag1, Jag2 and Dll1 and we have revised Figure 6 accordingly. We hope the revision makes this clearer.

7) In the legend to Figure 6, the statement 'As a result, these cells do not respond to hair cell-inducing signals' seems to contradict the sentence at the beginning of the section describing panel B, which states that these cells are responding to hair cell-inducing signals, and have been shaded green to indicate this. If the sentence refers to the neighbouring cells, the 'these' is ambiguous, and the sentence does not follow logically from the previous one. If the authors mean that the green cells have responded to hair cell-inducing signals, but have not yet received instruction to undergo final differentiation into hair cells (as in C), this needs clarifying further.

We have modified Figure 6 in response to comments from the reviewers, and we have also revised the legend and description of the figure to make it clearer. We have also added another figure (Figure 7) to try and better explain the mutant phenotypes.

Reviewer #2:

The paper by Basch et al. is a very interesting analysis of the function of Fringe proteins in hair cell patterning in the ear. The main and certainly interesting aspect of the work is the disclosure of a very specific function of Fng proteins in cochlear development and the exploration of their association with Notch signalling. The experiments are extremely careful and beautifully presented, and there is no doubt about the consistency of the various phenotypes and treatments. Data are of high quality and involve a variety of approaches including lineage tracing, phenotypic analysis, organ culture, etc., all making it a robust piece of work. The authors propose an interesting model that involves Notch activity levels and cis- and trans- modes of ligand operation to accommodate the results and with those of the literature. However, it is here where the work requires some improvement.

1) Figure 1 shows that the expression of Lfng and Mfng are restricted to sensory patch. Although the expression patterns are overlapping, they are quite dissimilar. This suggests that modulation of Notch is different in the two cell types. Could it be that Lfng and Mfng have different effects on different ligands? Is it possible that Lfng maintains a low and constant Notch activity and Mfng favours Dll signalling?

To date, the only strong *quantitative* evidence for the function of Fringe modifications on Notch signaling comes from cell line culture systems, such as the one used by the Elowitz group in their 2014 *eLife* paper. In that paper, they found no difference between the effect of Lfng and Mfng on Notch signaling, although they did find that Rfng behaved a little differently to the other two Fringe proteins. Moreover, there are only a few documented cases – including our study – where multiple Fringe knockouts have a more severe phenotype than Lfng alone. So, while it is *possible* that Lfng and Mfng have different but related functions in the cochlea, the tools and markers we currently have at our disposal cannot reveal this.

No doubt about the specificity of the combination of Lfng+Mfng for the correct development of the inner row. However, the correlation with Notch signalling levels remains obscure. Lfng+Mfng phenotype gives an expanded inner cell row, and this is paralleled by the loss of Jag2, but also of Jag1. Jag1 is expressed all throughout prosensory development and in supporting cells, but Jag2 is expressed along with Dll1 only in hair cells. How this is matched?

Our claim in the manuscript is that the newly differentiating Atoh1+ progenitors at the edge of the organ of Corti signal to their neighbors in Kölliker’s organ and prevents them from adopting the same fate. Our data shows that reducing the amount of Notch signaling delivered by these Atoh1+ cells or received by the Kölliker’s organ cells by even a small amount causes the adjacent Kölliker’s organ cells to adopt an Atoh1+ progenitor fate. We show this small reduction can be achieved in multiple ways, including by removing one copy of Jag1. In these cases, the genes we are knocking out are expressed broadly, but the effects are confined to the future inner hair cell region. Our conclusion is that this region is most sensitive to changes in Notch signaling.

2) Lineage tracing. Those are interesting experiments and show that almost the complete sensory epithelium derives from Lfng expressing progenitors (probably also from Jag1-positive ones). However, genetic tracing shows the common origin of the sensory epithelium, but not much about the restricted expression. In this view, the genetic tracing of Manic cells would have been of interest, because it seems that the uniqueness of the effects may come from the co-expression.

We agree that Mfng-Cre mice would have been a more specific way to show that inner phalangeal cells and inner hair cells derived from a common Mfng-expressing progenitor. However, those mice are not available to our knowledge, and we instead used Lfng-CreER mice that were developed in our lab for other purposes (such as mapping the fates of the early Lfng+ neurosensory domain, and for modifying genes in postnatal Lfng+ supporting cells). Although the Lfng expression pattern is very dynamic, by labeling between E13-E14, we were able to demonstrate that Lfng is expressed in the early Atoh1+ progenitors that give rise to both inner phalangeal cells and inner hair cells as shown in Figure 2.

3) Fng LOF is mimicked by low Notch. That is the main line of the argument along the paper. But from here, I would expect the authors to consider how this could happen by showing the effects of Fng on ligands or on Notch activity would have been informative. Neither it is clear to me whether the levels of Notch affect the expression of modulators or vice versa. We do not really know how Notch operates outside lateral inhibition and all possibilities should remain open.

We have now included extra data in Figure 4 showing a decrease in Notch activity (as revealed by N1ICD staining) in the Lfng/Mfng mutants. It is of course possible that lowering Notch signaling or Fringe activity will have other effects on the system, but we are at present limited by our readouts – functional (numbers of hair cells and supporting cells) and molecular (N1ICD staining or expression of Hes/Hey genes).

4) As to the levels of Notch activation, indeed, the observations fit well with the Murata paper. But nevertheless, the different conditions generated by the experiments in the paper open some questions about what are the levels of Notch in the different regions of the sensory epithelium. In this sense, conditions are not strictly comparable. Mfng and Lfng show overlapping but different expression domains, and in MAM and Pofut CKO or DAPT experiments, the decrease of Notch is ubiquitous. Can this be assessed specifically for Fng mutants? Please comment on that.

As mentioned above, we have now included extra data in Figure 4 to show the lowering of Notch activity in the Lfng/Mfng mutants. We hope this addresses this point.

5) The data in the bar diagrams of Figure 5 indicate that there is an inverse correlation between Notch activity, Hes/Hey expression and the production of hair cells. And it shows the differential sensitivity of inner and outer hair cells to Notch. This is in good correspondence with MAMl and Pofut experiments. But what about in Fng mutants? It would have been interesting to see Hes/Hey levels in Fng mutants.

The available data on Hes/Hey gene expression in the cochlea at this stage suggests that Hes and Hey are expressed at low levels at the border of the future organ of Corti (see, for example the E14.5 data in Figure 1 of Tateya et al., 2011). Although this is entirely consistent with our conclusion that the levels of Notch signaling at this border are quite low and sensitive to small changes, it makes it very hard to detect quantitative changes, especially since in situ hybridization itself is not very quantitative.

6) Cis-inhibition: The experiments say little about cis-inhibition. "Lfng activity attenuates Jag1-Notch signaling between cells (Hicks et al. 2000; Rana and Haltiwanger 2011) and promotes Jag1-Notch cis-inhibition in the same cell, with the result…" I am not sure how the second sentence follows the first and, moreover, whether it is a general true at all. Is there a good evidence for cis-inhibition in ear sensory cells? The results from Daudet's lab indicate rather the opposite for Dll1.

The argument on cis-inhibition continues in the second and third paragraphs of the Discussion, quoting LeBon 2014 as if an ear paper: "we hypothesize that this column of cells only delivers a moderate level of Notch signaling, due to active cis-inhibition between the Notch ligands and Notch receptors that can occur in the same cell". However, the experiments do not provide any evidence of that occurring in the ear.

We have rewritten these parts of the Discussion to clarify that cis-inhibition is *predicted* to occur in the cochlea when Notch ligands and receptors are expressed in the same cell. As mentioned above, the majority of evidence for cis-inhibition comes from both mammalian cell lines and *Drosophila* genetics. The process of cis-inhibition cannot be visualized by a unique marker; rather it is inferred by a) evidence for co-expression of receptor and ligand in the same cell and b) evidence for intermediate levels of Notch signaling that can be increased or decreased by manipulating levels of receptor, ligand or Fringe proteins in the cell. The low level of Notch signaling seen in the E13/14 prosensory domain that expresses both Jag1 and Notch1 is thus consistent with the phenomenon of cis-inhibition. The Daudet paper of 2012 is based on over-expression experiments and it is possible that the residual Notch signaling was sufficient to induce the rather sensitive Hes5-dsEGFP reporter in their experiments.

In summary, the results are indeed sound and of general interest, but as to the model, it is unclear that the function of Fng on Notch and cis-inhibition are sufficiently substantiated. Leaving aside cis-inhibition, which actually I see out of the paper, the model still needs a link between Fng, ligands and Notch in the cochlea. Something that may give a hint of what are the effects of Fng on Notch, Notch ligands or Fng. The main evidence for the model is correlative, the parallel between phenotypes. The main question, in my view, is what is the connection between both.

We have now added data showing that Lfng;Mfng double mutants have significantly reduced Notch signaling at the time and place where the boundary of the organ of Corti forms. We hope this adds weight to the idea that Lfng and Mfng activity are together necessary to promote Notch signaling from the cells that transiently co-express them.

Reviewer #3:

The organ of Corti is characterized by strikingly precise rows of hair cells, although how this precision is established is not known. This manuscript by Basch et al., proposes a novel form of Notch signaling that sets up the boundary of the initial row of inner hair cells, in which Fringe proteins, modulators of Notch signaling, are an essential component. The authors show that Lfng and Mng are initially precisely positioned at the location of the first inner hair cell row. In their absence, multiple rows form, along with a duplication of supporting cells. By comparing the results of minor reductions in Notch signaling to more robust loss of Notch, the authors suggest that establishment of the initial row of hair cells is inherently different in that it requires lower levels of Notch signaling, and is therefore more susceptible to mild reductions in Notch signaling. Overall the data is well-performed and presented, and the authors analyze quite a number of different Notch mutants to determine differences. The authors put forward an interesting hypothesis as to why the hair cell genesis is initially restricted to a single row. However, it is not clearly established that a lower level of Notch signaling is required for boundary formation – while milder defects in Notch signaling may first lead to an expansion of the sensory domain, it may be that the non-sensory regions (or supporting cells outside the hair cell regions) are simply more sensitive to loss of Notch signaling, and thus boundary formation is lost first.

The authors suggest that cis-inhibition mediated by Lfng and Mfng expression results in moderate levels of Notch activation in nearby Kolliker's cells (Figure 6), and that this is important for setting up the boundary. However, in this case the prediction would be that deleting both fringes would lead to higher levels of Notch, as cis-inhibition would be largely relieved. However, this does not seem to be the case, as new hair cells develop, suggesting Notch activity is reduced in their absence. Since the model indicates that moderate levels of Notch activity are important for setting up this boundary, it would be important to show directly how levels of Notch are regulated at the boundary.

As mentioned above, we have now provided data showing that Lfng;Mfng double knockouts have reduced Notch signaling in the border region of the future organ of Corti.

*It is not clear that severe loss of Notch signaling shows an inherently different phenotype than milder forms. Is it not more likely that the severe loss obscures the initial milder phenotype? After all, there is a dramatic loss of boundary formation in the severe cases of loss of Notch signaling. While additional supporting cells are present in the milder cases, these may be induced by the excess hair cells, which themselves also convert to hair cells in the case of more severe reductions in Notch.*

This is exactly what we believe is happening in our Notch1 mutants and previously published Notch1 mutants – there is a duplication of inner hair cell and inner phalangeal cell progenitors at the boundary of the organ of Corti, but the loss of Notch1 causes the inner phalangeal cell progenitors to form hair cells. We have stated this more explicitly in the revision.

If the regulation of inner hair cell formation to a single row is not via lateral inhibition, what type of signaling is it? It would seem that even at the proposed lower levels the cell expressing the ligand is inhibiting nearby cells (in Kolliker's organ) from adopting the hair cell fate, and therefore this would fit the definition of lateral inhibition (see Figure 6).

We do believe the signal is inhibitory, but it is clearly different from conventional Notch mediated lateral inhibition, both qualitatively (we see a different loss-of-function phenotype) and quantitatively (it is sensitive to changes in Notch signaling that do not affect the choice between hair cells and supporting cells). We felt that referring to this phenomenon was lateral inhibition would be confusing.

The authors show that Lfng likely initially marks the inner hair cell using fate-mapping, although given that they have reporters for both Lfng and Atoh1-it seems relatively straightforward to show that these both overlap during differentiation, further strengthening the hypothesis that Lfng and Mfng mark the inner hair cell boundary. It would also be interesting to look at the onset of these factors-Does Lfng mark the inner hair cell prior to Atoh1? Or vice versa?

The reason we chose to use the Lfng-CreER mice to show that inner hair cells derive from Lfng-expressing progenitors rather than other methods is that a) fluorescent protein reporters can persist after the gene is switched off; b) this is also true of protein reporters such as Atoh1-GFP fusion mice we have used in the past and c) in situ hybridization probes (either fluorescent or DIG-labeled) each have different sensitivities (as can be seen in Figure 1—figure supplement 1), making it hard to determine the precise temporal order of expression if expression patterns are changing rapidly as they do in the developing cochlea. Our expression and fate-mapping data in Figure 1 and Figure 2 show that Lfng is expressed broadly in Kölliker’s organ before becoming restricted to the boundary region. Therefore, it is likely that Lfng is expressed in hair cell progenitors first, but since we don’t have a unique marker of hair cell progenitors that is expressed before Atoh1, we can’t address this question. We recently showed that Mfng is a candidate Atoh1 target gene (Cai et al., 2015), but this has not been demonstrated directly.

The suggestion that Lfng modulates Jag1 levels initially (Figure 6) has not been demonstrated previously-is there any evidence of this from the Lfng loss of function? If not this part of the model should be modified.

We were not suggesting that Lfng modulates Jag1 *levels*, but rather Jag1 *activity*, which has been well-documented in the past in vivo and in vitro. We have now tried to clarify this in the text.